

**Processes controlling the seasonal variations of [210]Pb and**
**[7]Be at the Mt. Cimone WMO-GAW global station, Italy: A**
**model analysis**
Erika Brattich[1], Hongyu Liu[2], Laura Tositti[1], David B. Considine[3], and James H.
Crawford[4]
[1] Department of Chemistry "G Ciamician", Alma Mater Studiorum University of
Bologna, Bologna (BO), 40126, Italy
[2] National Institute of Aerospace, Hampton, Virginia, Virginia, VA 23681, USA
[3] NASA Headquarters, Washington, DC 20546, USA
[4] NASA Langley Research Center, Hampton, Virginia, VA 23681, USA
**Correspondence to**: Hongyu Liu (hongyu.liu-1@nasa.gov)
**Abstract.** We apply the Global Modeling Initiative (GMI) chemistry and transport model
driven by the NASA's MERRA assimilated meteorological data to simulate the seasonal
variations of two radionuclide aerosol tracers (terrigenous [210]Pb and cosmogenic [7]Be) at the
WMO-GAW station of Mt. Cimone (44°12' N, 10°42' E, 2165 m asl, Italy), which is
representative of free-tropospheric conditions most of the year, during 2005 with an aim to
understand the roles of transport and precipitation scavenging processes in controlling their
seasonality. The total precipitation field in the MERRA data set is evaluated with the Global
Precipitation Climatology project (GPCP) observations, and a generally good agreement is
found. The model reproduces reasonably the observed seasonal pattern of [210]Pb concentrations,
characterized by a wintertime minimum due to lower [222]Rn emissions and weaker uplift from





the boundary layer and summertime maxima resulting from strong convection over the
continent. The observed seasonal behavior of $^7$Be concentrations shows a winter minimum, a
summer maximum, and a secondary spring maximum. The model captures the observed $^7$Be
pattern in winter-spring, which is linked to the larger stratospheric influence during spring.
However, the model tends to underestimate the observed $^7$Be concentrations in summer,
partially due to the sensitivity to spatial sampling in the model. Model sensitivity experiments
indicate a dominant role of precipitation scavenging (versus dry deposition and convection) in
controlling the seasonality of $^{210}$Pb and $^7$Be concentrations at Mt. Cimone.
**1 Introduction**

11        The use of atmospheric radionuclides to understand atmospheric dynamics, pollution

transport and removal processes has a long history (e.g., Junge, 1963; Reiter et al., 1971;
Gäggeler, 1995; Arimoto et al., 1999; Turekian and Graustein, 2003; WMO-GAW, 2004; Dibb,
2007; Rastogi and Sarin, 2008; Froehlich and Masarik, 2010; Lozano et al., 2012). It has been
recognized that natural radionuclides are useful in a global monitoring network for atmospheric
composition to support global climate change and air quality research, and therefore they are
measured at many of the regional, global and contributing-partner stations in the Global
Atmosphere Watch (GAW) network of the World Meteorological Organization (WMO)
(WMO-GAW, 2004). In particular, terrigenous $^{210}$Pb and cosmogenic $^7$Be natural radionuclides
are helpful in the understanding of the roles of transport and/or scavenging in controlling the
behaviors of radiatively active trace gases and aerosols (Balkanski et al., 1993; Koch et al.,
1996), as well as their anthropogenic (vs. natural) origin (e.g., Graustein and Turekian, 1996;
Arimoto et al., 1999; Liu et al., 2004; Cuevas et al., 2013). They are routinely monitored at
WMO-GAW stations around the world (Lee et al., 2004). Although $^{210}$Pb and $^7$Be have long
(1998-2011) been measured at the Global WMO-GAW station of Mt. Cimone (Italy), their

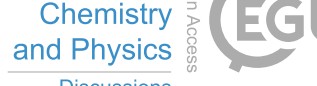
seasonal behavior has not been thoroughly elucidated (Lee et al., 2007; Tositti et al., 2014).
Here we apply a state-of-the-art global chemistry and transport model (CTM) to the simulation
of $^{210}$Pb and $^{7}$Be, with an objective to better understand the roles of transport and precipitation
scavenging processes in controlling their seasonal variations at Mt. Cimone.

5        Because of their contrasting natural origins, $^{210}$Pb and $^{7}$Be have been used as a pair to

study the vertical transport and scavenging of aerosols (Koch et al., 1996). $^{210}$Pb (half-life $\tau_{1/2}$
= 22.3 years) is the decay daughter of $^{222}$Rn ($\tau_{1/2}$ = 3.8 days), which is emitted from soils by
decay of $^{226}$Ra. The oceanic input of $^{222}$Rn is about two orders of magnitude less than the
continental input and, because of the continental origin of $^{222}$Rn, $^{210}$Pb is considered as a tracer
of air masses with continental origin (Baskaran, 2011). $^{7}$Be ($\tau_{1/2}$ = 53.3 days) is a cosmogenic
radionuclide generated by cosmic ray spallation reactions with nitrogen and oxygen (Lal et al.,
1958). Most (~67%) of $^{7}$Be is produced in the stratosphere and the remaining (~33%) is
generated in the troposphere, particularly in the upper troposphere (Johnson and Viezee, 1981;
Usoskin and Kovaltsov, 2008). $^{7}$Be is thus considered a tracer of stratospheric influence
(Viezee and Singh, 1980; Dibb et al., 1992, 1994; Liu et al., 2004, 2016) and subsidence (Feely
et al., 1989; Koch et al., 1996; Liu et al., 2004). Once produced, both radionuclides rapidly
attach onto aerosol particles in the fine fraction (Papastefanou and Ioannidou, 1995; Winkler
et al., 1998; Gaffney et al., 2004; Ioannidou et al., 2005), and are removed from the atmosphere
mainly by wet and secondarily dry deposition (Kulan et al., 2006). The concentrations of these
radionuclides in surface air thus depend on their sources, transport, wet and dry removal, and
radioactive decay (in the case of $^{7}$Be). Rainfall scavenging processes are generally more
effective on $^{210}$Pb than on $^{7}$Be concentrations (Koch et al., 1996; Caillet et al., 2001; Likuku,
2006; Dueñas et al., 2009; Lozano et al., 2012).





Observational studies have previously been conducted to examine the factors influencing
surface $^{210}$Pb and $^{7}$Be concentrations in Europe, the Middle East and North Africa. Different
synoptic and mesoscale patterns are associated with the ranges of $^{210}$Pb and $^{7}$Be activity
concentrations (Lozano et al., 2012, 2013). In southwestern Spain (El Arenosillo), for instance,
low $^{210}$Pb values are strongly linked to air masses from the Atlantic Ocean, whereas the highest
values are associated with air masses clearly under the influence of continents, such as the
Iberian Peninsula and North of Africa (Lozano et al., 2012). As for $^{7}$Be, the highest $^{7}$Be activity
concentrations over southwestern Iberian Peninsula are related with the arrival of air masses
from middle latitudes, and in particular from the Canary Islands, western Mediterranean Basin
and the north of Africa (Dueñas et al., 2011; Lozano et al., 2012).
With respect to $^{210}$Pb and $^{7}$Be spatial variability, $^{210}$Pb concentrations in surface air are
strongly dependent on whether it is located over land or ocean, whereas $^{7}$Be concentration is
mainly latitudinally dependent, due to their different production mechanisms. Generally
speaking, in the Northern Hemisphere higher $^{7}$Be concentrations are present at middle latitudes
(20-50° N), because of the mixing of stratospheric air into the upper troposphere along the
tropopause discontinuity in mid-latitude regions and subsequent convective mixing within the
troposphere, which brings $^{7}$Be-rich air masses into the planetary boundary layer and to the
earth's surface (Kulan et al., 2006). Lower $^{7}$Be concentrations are towards the pole and towards
the equator (Kulan et al., 2006; Steinmann et al., 2013).
Many studies examined the seasonal behavior of $^{210}$Pb and $^{7}$Be at European mid-latitude
surface sites (e.g., Cannizzaro et al., 1999; Ioannidou et al., 2005; Daish et al., 2005; Todorovic
et al., 2005; Likuku, 2006; Dueñas et al., 2009; Pham et al., 2011; Carvalho et al., 2013;
Steinmann et al., 2013). High levels of $^{210}$Pb during summer and low levels in winter were
found, reflecting the differing rates of $^{222}$Rn emanation from soil above the European land mass



during winter (wet or snow covered soil) and summer (dry soil) (Hötzl and Winkler, 1987;
Caillet et al., 2001; Daish et al., 2005; Ioannidou et al., 2005). At low-elevation sites, monthly
$^7$Be averages are characterized by a well-defined annual cycle with lower values during winter
and higher values during summer. Generally, the increase of $^7$Be in ground level air from March
to May is ascribed to the more efficient and higher frequency stratosphere- troposphere
exchange (STE), whereas the further increase of $^7$Be during summer is due to the stronger
convective mixing and higher tropopause (Ioannidou et al., 2014). The higher tropopause
height is associated with anticyclonic conditions, which results in downward transport from the
upper troposphere and reduced wet scavenging during these conditions (Gerasopoulos et al.,
2001, 2005; Ioannidou et al., 2014). In fact, compensating subsidence associated with
convective mixing enhances downward transport of $^7$Be from the upper troposphere (rather
than direct input of stratospheric air) down to the lower troposphere and ground level (Zanis et
al., 1999; Gerasopoulos et al., 2001, 2005; Ioannidou et al., 2005; Likuku et al., 2006;
Steinmann et al., 2013).
High-elevation sites such as Jungfraujoch (Switerland), Zugspitze (Germany), and Mt.
Cimone (Italy), typically lying above the planetary boundary layer (PBL), are characterized by
lower $^{210}$Pb concentrations and higher $^7$Be due to direct influences of air masses from the free
troposphere (Zanis et al., 2000). The observed seasonal $^{210}$Pb pattern at the high altitude sites
of Puy de Dome (1465 m asl, France) and Opme (660 m asl, France) is characterized by
maximum concentrations in spring and autumn and minimum concentrations in winter. This is
due to higher radon emissions during the dry season (summer) than during the wet season
(winter), and lower PBL height during winter (Bourcier et al., 2011). The latter results in
weaker upward transport of $^{222}$Rn and $^{210}$Pb at high-altitude sites. Similar to low-elevation sites,
higher $^7$Be values are observed in summer due to convection-forced exchange with the upper
troposphere and to the higher tropopause height that leads to more efficient vertical transport





from the upper to lower troposphere (Reiter et al., 1983; Gerasopoulos et al., 2001; Bourcier et
al., 2011). At high-altitude sites a secondary maximum of [7]Be during cold months (December-
March) is generally observed and attributed to the increase in stratosphere-to-troposphere
events during this season (e.g., James et al., 2003; Stohl et al., 2003; Trickl et al., 2010). The
higher frequency of rapid subsidence in winter at Northern Hemisphere mid-latitudes can be
ascribed to the intensity of baroclinic systems, which is greatest in the wintertime. In fact, well-
developed tropopause folds and rapid deep intrusions are most likely to occur in the wake of
intense cyclogenesis, usually limited to the wintertime storm track regions (James et al., 2003).

9         Numerical models have been used to analyze $^{210}$Pb and [7]Be observations at high-elevation

sites. 1-D model simulations of surface [7]Be showed higher concentrations at high-elevation
sites (Jasiulionis and Wershofen, 2005; Simon et al., 2009), but also suggested that the diffusion
of [7]Be was affected by the seasonal variation of meteorological conditions. Balkanski et al.
(1993) examined the transport of $^{210}$Pb in a global 3-D model and reported a weak decrease of
$^{210}$Pb concentrations between the continental mixed layer and the free troposphere: simulated
concentrations at 6-km altitude were about 50% of those in the continental mixed-layer over
much of the Northern Hemisphere in summer, and over large areas of the tropics year around,
a result consistent with the few observations available for the free troposphere at that time
(Moore et al., 1973). Rehfeld and Heimann (1995) compared the 3-D model simulated seasonal
pattern of surface $^{210}$Pb and [7]Be concentrations with the observations at several sites in both
hemispheres. At Mauna Loa (19.47°N, 155.6°W, 3400 m asl, Hawaii) $^{210}$Pb seasonality was
characterized by high concentrations in spring and summer and lower ones in winter, as
opposed to the seasonal pattern found at higher latitudes, where the $^{210}$Pb maximum
concentrations in winter are attributed to the advective transport of $^{210}$Pb aerosols from mid-
latitudes. This behavior is due to the elevation of the site, representative of the conditions of
the free troposphere rather than those of the PBL. As for [7]Be, the comparison between the





model and the observations at Rexburg (43.8°N, 111.83°W, 1483 m asl, USA) showed
systematically lower model values, due to the much higher precipitation rates in the model.

3        Previous studies have examined surface $^7$Be observations at Mt. Cimone with respect to

the role of STE in surface ozone increases (Bonasoni et al., 1999, 2000ab; Cristofanelli et al.,
2003, 2006, 2009a, 2015; Lee et al., 2007) within the framework of European projects such as
VOTALP (Vertical Ozone Transport in the Alps) and STACCATO (influence of Stratosphere-
Troposphere exchange in A Changing Climate on Atmospheric Transport and Oxidation
capacity). These studies led to the assessment of a higher incidence of STE events during the
period from October to February relative to the warm season, when thermal convection and the
rising of the tropopause promote vertical mixing, which acts as a confounding factor in STE
detection. Lee et al. (2007) and Tositti et al. (2014) reported the seasonal patterns and frequency
distributions of $^{210}$Pb and $^7$Be measured at Mt. Cimone, and highlighted higher concentrations
of both radionuclides in the summertime due to the higher mixing height and horizontal
transport by regional airflows. During winter, a general increase in $^7$Be is associated with a
decrease in $^{210}$Pb, due to the dominating effect of STE and subsidence in the free troposphere.
At the time of this work, no model analyses of $^{210}$Pb and $^7$Be observations at the site have been
conducted.

18       In this paper, we conduct simulations of $^{210}$Pb and $^7$Be at Mt. Cimone with a state-of- the-

art global 3-D chemistry and transport model (GMI CTM) driven by assimilated
meteorological fields for the year of 2005. Our objectives are a better elucidation of the
seasonal variations of $^{210}$Pb and $^7$Be concentrations and an improved understanding of the roles
of transport and precipitation scavenging processes in their seasonalities at Mt. Cimone.

23       The remainder of this paper is organized as follows. Section 2 describes the measurement

site, the radioactivity measurements at Mt. Cimone, and the GMI CTM. Section 3 evaluates



the model performance in reproducing the observed wind and precipitation fields. Section 4
evaluates the seasonal $^{210}$Pb and $^{7}$Be concentrations in the model with those observed. Section
5 examines the sources and seasonal variations in the simulated radionuclide activities,
followed by summary and conclusions in section 6.
**2 Data and Methods**
**2.1 Radionuclide Measurements at Mt. Cimone**
Mt. Cimone station (44°12' N, 10°42' E, 2165 m asl) is a global WMO-GAW station
managed by the Meteorological Office of the Italian Air Force, which hosts the research
platform "Ottavio Vittori" of the Institute of Atmospheric and Climate Science of the National
Council of Research (ISAC-CNR). The station is located on top of the highest peak of the
Italian northern Apennines, with a 360° free horizon and an elevation such that the station lies
above the PBL during most of the year: the Mt. Cimone measurements are considered
representative of the southern Europe/Mediterranean free troposphere (Bonasoni et al., 2000a;
Fischer et al., 2003; Cristofanelli et al., 2007), although during the warmer months an influence
of PBL air can be detected due both to convective processes and mountain/valley breeze
regimes (Fischer et al., 2003; van Dingenen et al., 2005; Tositti et al., 2013). Note in this
framework that southern Europe and Mediterranean basin are considered as a hot-spot region
in terms of both climate change (e.g., Forster et al., 2007) and air quality (Monks et al., 2009),
as well as a major crossroad of different air mass transport processes (Li et al., 2001; Lelieveld
et al., 2002; Millàn et al., 2006; Duncan et al., 2008; Tositti et al., 2013).
At Mt. Cimone station, $^{210}$Pb, $^{7}$Be, and aerosol mass load in the form of $PM_{10}$ have been
regularly measured in the period of 1998-2011 with a Thermo-Environmental $PM_{10}$ high-
volume sampler. $PM_{10}$ is sampled on rectangular glass fiber filters (Whatman, 20.3 cm × 25.4
cm, with an effective exposure area of about 407 cm$^2$), which were manually changed every 2-



3 days, depending on weather conditions, failures of the sampling equipment and/or of the
power supply and personnel on site. The average flow rate was about 1.13 $m^3$ $min^{-1}$ at standard
temperature and pressure (STP), with an average volume of air collected on each filter equal
to 3000-4000 $m^3$ (about 48 hours of sampling, 115-175 samples per year).
Airborne radionuclides travel attached to particulate matters, and as a consequence of
their physical origin, tend to populate the fine fraction (<1.0 μm) (Winkler et al., 1998; Gaffney
et al., 2004). The $PM_{10}$ samples were subjected to non-destructive high-resolution γ-
spectrometry for the determination of airborne radiotracers $^{210}$Pb and $^{7}$Be. The characteristics
of the two Hyper Pure Germanium crystal detectors (HPGe) detectors are as follows: one p-
type coaxial detector by Ortec/Ametek with a relative efficiency of 32.5% and FWHM 1.8 keV
at 1332 keV and one planar DSG detector with an active surface of 1500 $mm^2$ and FWHM 0.73
keV at 122 keV, for higher and lower energy ranges (100-2000 keV and 0-900 keV),
respectively.
Spectra were accumulated for at least one day to optimize peak analysis and then
processed with a specific software package (GammaVision-32, version 6.07, Ortec). Efficiency
calibration was determined on both detectors with a blank glass fiber filter traced with
accurately weighted aliquots of a standard solution of mixed radionuclides (QCY48,
Amersham) supplemented with $^{210}$Pb, homogeneously dispersed dropwise over the filter
surface. Once dried under a hood under ambient conditions, the calibration filter was folded
into a polystyrene container in the same geometry as the unknown samples. Quantitative
analysis on samples was carried out by subtracting the spectrum of a blank filter in the same
geometry, while uncertainty on peaks (k = 1, 68% level of confidence) was calculated
propagating the combined error over the efficiency fit previously determined with the counting
error. Minimum detectable activity was calculated making use of the traditional ORTEC
method (ORTEC, 2003) with a peak cut-off limit of 40%. Activity data was corrected to the



midpoint of the time interval of collection and for the decay during spectrum acquisition. For
our analysis, we used monthly averages of $^{210}$Pb and $^{7}$Be data at Mt. Cimone in 2005.
**2.2 GMI Model**

4        The Global Modeling Initiative (GMI, http://gmi.gsfc.nasa.gov) is a NASA-funded

project aiming at improving assessments of anthropogenic perturbations to the Earth system;
in this framework a CTM appropriate for stratospheric assessments was developed (Rotman et
al., 2001). It was firstly used to evaluate the potential effects of stratospheric aircraft on the
global stratosphere (Kinnison et al., 2001) and on the Antarctic lower stratosphere (Considine
et al., 2000). The recent version of the GMI CTM includes a full treatment of both stratospheric
and tropospheric photochemical and physical processes and is also capable of simulating
atmospheric radionuclides $^{222}$Rn, $^{210}$Pb, $^{7}$Be, and $^{10}$Be throughout the troposphere and
stratosphere (Considine et al., 2004, 2005; Rodriguez et al., 2004; Liu et al., 2016). Details of
the model are described in Duncan et al. (2007, 2008), Strahan et al. (2007), and Considine et
al. (2008).

15       In this work a version of the GMI model with the same basic structure as described by

Considine et al. (2005) and Liu et al. (2016) was used, including parameterizations of the
important tropospheric physical processes such as convection, wet scavenging, dry deposition
and planetary boundary layer mixing. Meteorological data used to drive the CTM, e.g.,
horizontal winds, convective mass fluxes and precipitation fields, are the Modern-Era
Retrospective analysis for Research and Applications (MERRA) assimilated data set from the
NASA Global Modeling and Assimilation Office (GMAO) (Rienecker et al., 2011).

22       The flux-form semi-Lagrangian advection scheme and a convective transport algorithm

from the CONVTRAN routine in NCAR CCM3 physics package are used in the model. The
wet deposition scheme is that of Liu et al. (2001): it includes scavenging in wet convective
updrafts, and first-order rainout and washout from both convective anvils and large-scale





precipitations. The gravitational settling effect of cloud ice particles included in Liu et al.
(2001) is not considered here. Dry deposition of aerosols is computed using the resistance-in-
series approach. For the simulations of radionuclides, each simulation was run for six years,
recycling the meteorological data for each year of the simulation, to equilibrate the lower
stratosphere as well as the troposphere (Liu et al., 2001). The sixth year output was used for
analysis.

7        A uniform $^{222}$Rn emission of 1.0 atom cm$^{-2}$ s$^{-1}$ from land under nonfreezing conditions is

assumed (Liu et al., 2001). Following Jacob and Prather (1990), the flux is reduced by a factor
of 3 under freezing conditions. The flux from oceans and ice is null. Although a large variability
of $^{222}$Rn emission from land is observed, the above emission estimate is thought to be accurate
to within 25% globally (Turekian et al., 1977) and to within a factor of 2 regionally (Wilkening
et al., 1975; Schery et al., 1989; Graustein and Turekian, 1990; Nazaroff, 1992; Liu et al.,

13   2001).

14       Following Brost et al. (1991) and Koch et al. (1996), we used the Lal and Peters (1967)

$^{7}$Be source for 1958 (solar maximum year), as it best simulated stratospheric $^{7}$Be concentrations
measured from aircraft (Liu et al., 2001). No interannual variability in the $^{7}$Be source is
considered in the model (Liu et al., 2001). This may lead to an underestimate of tropospheric
$^{7}$Be concentrations, especially at high latitudes during a solar minimum (or near minimum)
year. Lal and Peters (1967) reported that the relative amplitude of the $^{7}$Be production rate over
a 11-year solar cycle is about 13% below 300 hPa at latitudes above 45 degree.

21       Because of the coarse horizontal resolution of the model (2° latitude by 2.5° longitude),

the model representation of the topography at the site is poor. The elevation of Mt. Cimone in
the model is only 298 m, whereas in reality the mountain is 2165 m (asl) high (Figure 1). For
this reason, the model output was not sampled at ground level, but at the gridbox corresponding
to the elevation of the site. In order to see the sensitivity of model-observation comparisons to





spatial sampling, the model was sampled not only for the grid corresponding to the latitude and
longitude of Mt. Cimone, but also for the 8 adjacent grids. To better understand the sources
and seasonality of radiotracers in the model, we examine model output not only for $^{210}$Pb, $^{7}$Be
and their ratio $^{7}$Be/$^{210}$Pb (an indicator of vertical transport [Koch et al., 1996]), which can be
directly compared to the measurements taken at Mt. Cimone, but also for other radiotracers and
quantities, e.g., $^{222}$Rn, and $^{10}$Be/$^{7}$Be (a STE tracer [Zanis et al., 2003]).

7       Year 2005 was chosen for analysis because of the availability of the observational data

and model output at the time of this work. As discussed later, the seasonal behavior of $^{210}$Pb
and $^{7}$Be radionuclides during year 2005 was "typical" for Mt. Cimone. Monthly averages of
$^{210}$Pb and $^{7}$Be data at Mt. Cimone were calculated for comparison with model results. To better
compare the seasonalities of $^{210}$Pb and $^{7}$Be between the model and the observations, the
monthly percentage deviations from the annual mean concentration were also calculated.
**3 Seasonal Variations of Transport and Precipitation at Mt. Cimone: Observations vs.**
**Model Simulations**
Mt. Cimone is the windiest meteorological station in Italy and the prevailing local winds
blow from S-SW and N-NE directions (Ciattaglia, 1983; Ciattaglia et al., 1987; Colombo et al.,
2000). The wind observations at Mt. Cimone during the period of 1998-2011, when
radionuclide measurements were performed at the station (Tositti et al., 2014), agree with the
climatology of local wind intensity and direction during the period of 1946-1999 as reported
by the Italian Air Force (Colombo et al., 2000). N-NE directions are more significant during
the cold period, and fluxes from SW are more typical of the warm period. While winds blowing
from the S-SW sector generate a sea air inflow, a continental air inflow is observed when winds
come from the N-NE sector (Ciattaglia et al., 1987).
However, when considering the lifetimes of $^{210}$Pb (about one week) and $^{7}$Be (about three
weeks) aerosols (Liu et al., 2001), it is apparent that the regional and long-range transport has



a much more important role than local transport. On a large scale, about 70% of background
air masses reaching Mt. Cimone in the period of 1996-1998 came from Atlantic and Arctic
areas, with a smaller contribution from the Mediterranean Basin and the eastern area, as
estimated by Bonasoni et al. (2000). A more recent and extended study of advection patterns
at Mt. Cimone (Brattich E. et al., "Advection patterns at the WMO-GAW station of Mt.
Cimone: seasonality, trends, and influence on atmospheric composition", manuscript in
preparation, 2016), analyzing clusters of 4-day kinematic back-trajectories calculated for the
period of 1998-2011 with the HYSPLIT (HYbrid Single-Particle Lagrangian Integrated
Trajectory) model driven by the NCEP/NCAR (National Center for Environmental
Prediction/National Center for Atmospheric Research) meteorological reanalysis, shows that
the air masses advected to Mt. Cimone (55%) arrive from the Western-Atlantic-North America
sector, while the remaining air masses (from the Arctic, Eastern and Mediterranean Basin-
Northern Africa) together represent 45% of trajectories. Seasonal transport to Mt. Cimone in
the model is shown in Figure 2, representing winds at the elevation of Mt. Cimone (winds are
weaker at the model bottom layer). In agreement with the description of advection patterns at
the site, prevailing model winds (Figure 2) blow from the western-Atlantic sector. Slow
summer winds suggest the stronger influence of regional/local transport at Mt. Cimone during
the period (e.g., Lee et al., 2007; Marinoni et al., 2008; Tositti et al., 2013, 2014; Brattich et
al., 2015).
In the model Mt. Cimone appears to be in a location where there is a large horizontal
gradient of wind (transport). Long-range transport from Western Europe, North America and
Arctic region prevail during the cold period, while regional transport appears more important
in summer. The model is able to capture relevant features of pressure systems and seasonal
circulation patterns of the North Atlantic/Mediterranean/African region, such as the semi-
permanent high pressure system located in the North Atlantic with different positions during





different seasons (Bermuda/Azores high), a semi-permanent system of high pressure centered
in northeastern Siberia during the colder half of the year (Siberian high), and the ITCZ in the
summer/autumn season. However, due to the coarse resolution of the global meteorological
reanalysis that we use to construct the model winds, the more than 50 local-scale wind systems
present in the Mediterranean and surrounding regions are not resolved (Burlando, 2009). In
northern Europe, in fact, there are approximately two main states for the atmosphere, the
westerly or zonal flows modulated by the advection of Atlantic lows, and the long-lived
blocking anticyclonic configurations over North Sea or Scandinavia (easterly) (Burlando et al.,

9    2008).

In the Mediterranean region, the main cyclones during winter are essentially sub-synoptic
lows triggered by the major North-Atlantic synoptic systems affected by the local topography
of the Northern Mediterranean coast (Trigo et al., 2002), whereas in summer cyclones develop
because of thermal effects, orography (e.g., the Atlas Mountains), and increase in low-level
thermal gradients (Trigo et al., 2002; Campins et al., 2006). Again, due to the coarse resolution
of the meteorological data we use, these sub-synoptic processes are not resolved. For instance,
North-African lows and Sahara depressions (also referred to as Atlas lee depressions) and the
resulting S-SW wind (Sirocco) (Reiter, 1975), potentially linked to $^{210}$Pb variations at Mt.
Cimone, appear to be an important feature missing in the degraded MERRA data, where they
appear only during October/November.
We evaluate the MERRA precipitation with those from the GPCP (Global Precipitation
Climatology Project, http://www.gewex.org/gpcp.html) satellite and surface observations in
2005. Figure 3 shows the MERRA and GPCP monthly precipitation for the region defined by
0-75°N and 90°W – 90°E. A good agreement between the MERRA and the GPCP
precipitations averaged over the region was found. In particular, summer precipitation patterns
are very similar. The geographical distribution of precipitation in MERRA shows some



important features in agreement with the observed climatology precipitations: the desert
climate in North Africa with very low precipitation all year long, the ITCZ with high
precipitation during the summer/autumn seasons, the North Atlantic region with high
precipitation especially during the winter and autumn seasons, and Europe where the seasonal
pattern of precipitation is similar to that in the North Atlantic region, but precipitation is lower.
Figure 4 shows the comparison of the GPCP and MERRA precipitation seasonality at Mt.
Cimone. Since Mt. Cimone is located in a region with a large horizontal gradient in
precipitation, we also show in the figure the comparisons for three adjacent gridboxes. The
agreement between the MERRA and GPCP precipitation seasonality is reasonable, with the
squared correlation coefficient $R^2$ varying between 0.56 (at the grid to the northwest of "ij")
and 0.89 (at the grid to the southeast of "ij"). Large differences between the MERRA
precipitation and that locally observed at the station are instead present (not shown): in
particular, the MERRA precipitation is larger during winter-autumn, while it is much more
similar to that observed during spring-summer. This difference may very well reflect again the
fact that the observed surface precipitation is localized, whereas the satellite and MERRA
precipitations correspond to a much larger scale (about 200 km). Moreover, as Colombo et al.
(2000) previously pointed out, different from the surrounding area where the climate is defined
as temperate-continental, the climate at the mountaintop is classified as alpine because of the
high elevation. In fact, in agreement with the GPCP precipitation in 2005, the observed
climatology in the region shows maximum during November (secondary maximum in spring)
and absolute minimum in July (secondary minimum in January), whereas on the top of the
mountain the precipitation is maximal during summer. The MERRA precipitation shows
increased amounts during April and August-December, with minimum in June-July. As the
local precipitation at the site is important to the scavenging of radionuclide aerosol tracers, this
difference between the local and regional precipitation could contribute to any biases in our





simulations. However, as we will show below, the ratio $^{7}Be/^{210}Pb$ may cancel out the errors
associated to precipitation scavenging (Koch et al., 1996).

3         Low $^{210}Pb$ concentrations are seen over the Atlantic Ocean, due to the negligible

emissions of $^{222}Rn$ from the oceans and strong precipitation scavenging, and in northern and
western Europe especially during the cold season (Figure 2a). High $^{210}Pb$ concentrations appear
over the Sahara desert and North Africa, as a result of low precipitation in this area, and also
over the Middle East and South Asia. $^{210}Pb$ concentrations over southern Europe appear higher
during the transition seasons, especially fall, and peak during summer when the minimum
precipitation and slow winds from west are observed in the region. Low $^{7}Be$ concentrations are
simulated along the equator where convective scavenging is strongest (Figure 2b). High $^{7}Be$
concentrations are seen over the Sahara desert due to a combination of low precipitation and
subsidence in this region. Elevated values also occur over the Middle East, North America, and
Greenland. $^{7}Be$ concentrations over southern Europe appear higher during spring and peak
during winter, when model winds are stronger and transport $^{7}Be$ aerosols from North America
and Greenland regions where $^{7}Be$ production is highest (Beer et al., 2012).
**4 Seasonal Variations of $^{210}Pb$ and $^{7}Be$ at Mt. Cimone: Observations vs. Model**
**Simulations**

18        The seasonality and frequency distributions of $^{210}Pb$ and $^{7}Be$ concentrations measured at

the Mt. Cimone station were previously examined by Lee et al. (2007), while more recent
analyses of the 12-year record were presented in Tositti et al. (2014) and Brattich et al. (2015).
Generally, both radionuclides show a marked seasonal maximum in the summertime, a
behaviour shared by $PM_{10}$ (Tositti et al., 2013) and $O_3$ (Bonasoni et al., 2000b). $^{210}Pb$ summer
maximum is mainly due to the higher mixing height and enhanced uplift from the boundary
layer as a result of thermal convection. The seasonal fluctuation of $^{7}Be$ is more complex and
characterized by two relative maxima, one during the cold season associated with stratosphere-





to-troposphere transport, and the other during the warm season mainly associated with
tropospheric subsidence balancing lower-tropospheric air masses ascent occasionally
accompanied by STE (Tositti et al., 2014). The $^{210}$Pb and $^{7}$Be measurements in 2005 are
consistent with this description (Figure 5): $^{210}$Pb concentrations are characterized by two
maxima during the warm period (July and September); $^{7}$Be concentrations are characterized by
one absolute maximum during summer (July) and one secondary maximum during spring
(March).

8        Figure 5 (ab) compares the simulated monthly $^{210}$Pb and $^{7}$Be activities with the

observations at Mt. Cimone in 2005. The comparisons for the monthly percentage deviations
from the annual mean concentration are available as Supplementary Information (hereafter SI,
SI Figures 1-2). The seasonality of $^{210}$Pb is well captured by the model. The model reproduces
the presence of two seasonal maxima in the $^{210}$Pb observations, with the maximum observed in
July shifted to June in the simulation. The squared correlation coefficient $R^2$ between observed
and simulated $^{210}$Pb activities is equal to 0.83 at the "ij" grid and varies between 0.42 and 0.82
for adjacent gridboxes (to the north and to the west of "ij", respectively), confirming the good
performance of the model in reproducing the $^{210}$Pb seasonal pattern.

17       As for $^{7}$Be, the model well captures the March maximum (i.e., secondary maximum in

the observations) and the general seasonal pattern during the cold and transition seasons.
However, during the warm period, the simulated $^{7}$Be concentrations are lower by a factor of 2
than the observed. A better agreement was found at some adjacent model gridboxes (e.g., "to
the south and to the southwest of "ij"; Figure 6 vs. Figure 5). The correlation between observed
and simulated monthly $^{7}$Be activities also increases from $R^2 = 0.03$ at "ij" to $R^2 = 0.11$-0.60 at
adjacent model gridboxes.
**5 Sources and Seasonality of $^{210}$Pb and $^{7}$Be at Mt. Cimone: A Model Analysis**





In this section, we quantify the sources of $^{210}$Pb and $^7$Be and determine the processes
governing their seasonality in the GMI model. Additional tracers as simulated by the model are
used to aid in the interpretation. Model sensitivity experiments are conducted to examine the
roles of transport and precipitation scavenging in the seasonality.
As discussed in Section 4, the model well reproduces the $^{210}$Pb seasonality, with
minimum in the cold period and maximum in the warm period. The $^{210}$Pb seasonality (Figure
5a) can be linked with the seasonal pattern of its precursor $^{222}$Rn (Figure 5c). It is seen that the
summer $^{210}$Pb maximum is due to stronger (thermal) convection, which uplifts more $^{222}$Rn out
of the boundary layer (e.g., Lee et al., 2007; Tositti et al., 2014; Brattich et al., 2015). This
uplift of $^{222}$Rn from the boundary layer is minimum in the cold period, and the minimal level
of $^{210}$Pb in this period can be considered representative of the free troposphere. The $^{210}$Pb
summer increase appears to be associated with short-range and regional transport, as suggested
by the model simulations (Figure 2a). As expected, long-range transport is more typical of the
winter/spring seasons because of stronger horizontal winds, while regional effects are more
important during summer when convection gets stronger.
In a similar manner, the source of the $^7$Be March maximum can be investigated with
model tracer simulations. Figure 5 (de) also shows the simulated seasonal patterns of the
$^{10}$Be/$^7$Be activity ratio and of the fraction of $^7$Be originating from the stratosphere (strat
$^7$Be/total $^7$Be). The simulated seasonal pattern of the $^{10}$Be/$^7$Be ratio is very similar to the
observations at Zugspitze (Germany, 2962 m asl) (Zanis et al., 2003), characterized by a not-
pronounced seasonal cycle with somewhat elevated ratios in February-April and June-July.
The usefulness of $^{10}$Be/$^7$Be ratio as a stratospheric tracer is due to the fact that both $^{10}$Be and
$^7$Be cosmogenic radionuclides attach to the same aerosols and share therefore the same removal
mechanism. Moreover, due to the much longer physical half-life of $^{10}$Be ($\tau_{1/2} = 1.5 \times 10^6$ years)
compared to $^7$Be ($\tau_{1/2} = 53.3$ days), their concentration ratios in the stratosphere (about 3-4) are



much higher than in the troposphere (about 2 or even less) (Koch and Rind, 1998). The
simulated $^{10}Be/^7Be$ ratio behavior indicates that deep stratosphere-to-troposphere (STT) peaks
during winter, while shallower STT has a spring maximum, consistent with previous analyses
of stratospheric intrusions at Mt. Cimone (Cristofanelli et al., 2006, 2009), and more generally
with the climatology of stratosphere-troposphere exchange at the Northern Hemisphere mid-
latitudes (James et al., 2003). Altogether the simulated high strat $^7Be$/total $^7Be$, high $^7Be/^{210}Pb$
(Figure 7), and low $^{10}Be/^7Be$ ratios during December-January indicate strongest STE during
this period, followed by spring with slightly weaker stratospheric influence on surface $^7Be$.
However, the model tends to overestimate the observed $^7Be$ concentrations and $^7Be/^{210}Pb$ ratios
during December-February, suggesting that STE and/or subsidence in the model is likely too
fast in this region. As reported by Huang et al. (2013), a stronger net subsidence of air masses
to the surface could be due to unrealistic meteorological conditions (e.g., boundary layer
structure, wind fields, vertical mixing).

14   The use of the $^7Be$ production rate of Lal and Peters (1967) for a solar maximum year

(1958) may partly explain the lower annual mean $^7Be$ in the model (3.4 mBq m$^{-3}$ annual mean
at the "ij" grid) than in the observations (4.2 mBq m$^{-3}$). In fact, the sunspot number in 2005
(29.8) was quite low (slowly decreasing from 2000, a solar maximum year, and reaching
minimum in 2008), especially compared to the 1958 value of 184.8. Sunspot number data are
available from the World Data Center for the production, preservation and dissemination of the
international sunspot number (Sunspot Index and Long-term Solar Observation, SILSO, Royal
Observatory of Belgium, Brussels, http://sidc.oma.be/sunspot-data/).

22   During the winter period, associated with the simulated and observed $^7Be$ increases

(Figures 5-6), strong long-range transport was dominant in the European region (Figure 2b).
Transport from higher latitude regions (Arctic, northern Europe, and North America) appears
particularly important during this period (Figure 2b); such transport from high-latitude regions,





where the $^{7}$Be production rate is highest (Beer et al., 2012), has typically been observed during
STE events at Mt. Cimone in many studies (e.g., Bonasoni et al., 1999, 2000ab).

3          The discrepancy between the simulated and the observed $^{7}$Be concentrations during the

warm period is partly due to the sensitivity to spatial sampling in the model. As seen from the
map plots of $^{210}$Pb and $^{7}$Be concentrations at the elevation of Mt. Cimone (Figure 2), the
sampling site appears to be located in a region where the N-S gradient of concentrations is large
(especially for $^{7}$Be). An elevated gradient in the region surrounding Mt. Cimone was also seen
for winds, as transport plays a critical role in determining the distributions of these tracers. The
sensitivity to spatial sampling in the model is therefore ascribed to this observed strong gradient
in the N-S direction. In fact, while the grids to the south and southwest of "ij" are better for
summer $^{7}$Be comparisons (Figure 6), the grids to the northeast, north, and northwest of "ij" are
better for winter (not shown).

13          The model underestimate of $^{7}$Be levels in the warm months may also suggest the mixing

of air masses between the PBL and the lower free troposphere is likely too weak. Previous
observational analyses indicated that such mixing is higher in summer at Mt. Cimone due to
enhanced convection and mountain wind breeze (e.g., Fischer et al., 2003; Cristofanelli et al.,
2007). Weaker entrainment of free-tropospheric air into the PBL would result in lower $^{7}$Be
concentrations at the surface.

19          The model annual average biases are about 8% for $^{210}$Pb and about 19% for $^{7}$Be,

respectively. By contrast, the model average bias for $^{7}$Be/$^{210}$Pb ratios is about -13% (Figure 7).
The smaller model bias for $^{7}$Be/$^{210}$Pb ratios than for $^{7}$Be concentrations reflects the fact that the
ratio cancels out the errors in precipitation scavenging (Koch et al. 1996) that contribute to the
underestimate of $^{210}$Pb and $^{7}$Be activities. On the other hand, the negative model bias for the
$^{7}$Be/$^{210}$Pb ratio again points to weak downward mixing from the free troposphere.





If one compares the month-to-month variation of $^{210}$Pb and $^{7}$Be (Figures 5 and 6) and
precipitation in the model (Figure 4), the maxima/minima of precipitation appear to be in phase
with those of both radionuclides' activities. This reflects the effects of precipitation scavenging
on radionuclide aerosols.
We conducted model sensitivity experiments where convection (transport and
scavenging), wet scavenging due to both large-scale and convective precipitation, and dry
deposition processes are turned off, respectively, to examine the roles of these processes in
controlling the seasonality of $^{210}$Pb and $^{7}$Be at Mt. Cimone. Figure 8 shows the results for the
standard and sensitivity runs at the "grid to the south of "ij", for which the simulated tracer
seasonal variations are similar to those observed, while the monthly percentage deviations from
the annual mean concentrations are shown in SI Figure 3. Figures 9-12 show maps of simulated
changes in $^{210}$Pb and $^{7}$Be concentrations when convection or wet scavenging is turned off.
Turning off dry deposition does not significantly change the simulated $^{210}$Pb and $^{7}$Be
concentrations, partly due to sampling the higher vertical gridbox in the model (larger effects
are seen at the bottom model layer). Turning off convection (i.e., with neither convective
transport nor convective scavenging), the simulated $^{7}$Be seasonality also remains nearly the
same. This suggests the compensating effects between subsidence (increasing $^{7}$Be) associated
with convective transport and scavenging (decreasing $^{7}$Be) due to convective precipitation. In
the case of $^{210}$Pb, turning off convection does not change the seasonal pattern but generally
results in larger $^{210}$Pb concentrations and particularly during summer/autumn when convective
transport is more important at the site. In fact, no convective transport of $^{222}$Rn (SI Figure 5)
results in less $^{222}$Rn (and $^{210}$Pb) being transported to the free troposphere, but also more $^{210}$Pb
available in PBL lifted to the free troposphere by large-scale vertical transport; on the other
hand, lack of convective scavenging of $^{210}$Pb increases its concentration in the free troposphere.
Turning off convection therefore results in an increase of $^{210}$Pb concentrations in the free





troposphere. Both surface $^{222}$Rn concentrations at the elevation of Mt. Cimone (SI Figure 4),
as well as a map of changes in $^{210}$Pb concentrations due to convection in the model (Figure 9)
show that convection in the region is more important during summer and autumn, but is not
negligible during spring, possibly due to thermal inertia.

5        The model run without scavenging suggests that, apart from downward transport from

the upper troposphere and lower stratosphere, wet scavenging is mainly responsible for the
seasonal variation of $^{7}$Be (Figure 8, bottom panel). None of our simulations is able to describe
the observed $^{7}$Be summertime peak, suggesting that the local circulation in this region with
complex topography may not be resolved by the coarse-resolution model. For $^{210}$Pb (Figure 8,
top panel), it appears that wet scavenging plays a more important role during August-December
than during January-July. This appears to be associated with the seasonality of precipitation,
which shows prolonged elevated values during August-December, as well as a maximum
during April, as previously discussed (Figure 5). A plot of changes in $^{210}$Pb concentrations due
to scavenging in the model (Figure 10) confirms that the scavenging effect is larger during fall
and, to a lesser extent, during summer. At Mt. Cimone, the scavenging effect is not minimal
during July (month of minimum precipitation, Figure 4), suggesting the influence of
precipitation scavenging elsewhere in the region on the site.
**6 Summary and Conclusions**

19       We have used a global 3-D model (GMI CTM) driven by the MERRA assimilated

meteorological data from NASA's GMAO to simulate the $^{210}$Pb and $^{7}$Be observations from the
Mt. Cimone (44°12' N, 10°42' E, 2165 m asl, Italy) WMO-GAW station in 2005. The two
natural atmospheric radionuclides originate from contrasting source regions (lower troposphere
and upper troposphere/lower stratosphere, respectively), attach to submicron particles, and are
removed from the troposphere mainly by wet deposition. Our objective was to examine the
roles of horizontal advection, vertical transport (large-scale and convection), and wet





scavenging in determining the seasonality of $^{210}$Pb and $^{7}$Be at Mt. Cimone. The observed $^{210}$Pb
concentrations are characterized by maxima in summer and minima during the cold period.
The seasonality of $^{7}$Be is more complex, with a major peak in summer, a secondary peak in
spring and a minimum in winter. This is the first modeling study of $^{210}$Pb and $^{7}$Be observations
at Mt. Cimone. This site is representative of free-tropospheric Southern Europe/Mediterranean
conditions most of the year, and as such the comparison between measurements and
simulations can serve as an indication of shortcomings in the model or in the meteorological
data.

9       Precipitation and wind fields are important to the model's performance in representing

the transport and scavenging processes. We evaluated the MERRA precipitation field used by
GMI CTM against the GPCP satellite and surface observations, and a generally good
agreement was found. The seasonality of precipitation at Mt. Cimone shows increased amounts
during April and the period of August-December, and minimum in June-July. The MERRA
assimilated winds at the low resolution version we used captured the main circulation patterns
(e.g., location of the Azores high pressure, location of the ITCZ) in the Northern Hemisphere.
However, some local-scale winds and pressure systems, which are important for transport to
the sampling site, were likely not well resolved at the coarse resolution we used. A general
good agreement was found between the MERRA assimilated wind fields and the main
advection patterns at the site (e.g., prevalence of long-range transport from Western Europe,
North America and Arctic region during the cold season, as opposed to the prevailing regional
transport during the warm season).
The model well reproduced the observed $^{210}$Pb seasonality: $^{210}$Pb maxima during the
warm period were attributed to the stronger (thermal) convection, which uplifts more $^{222}$Rn
(and $^{210}$Pb) from the boundary layer. The model is less successful in reproducing the observed
$^{7}$Be seasonality. $^{7}$Be was better represented during the cold period, while the observed summer



$^7$Be maximum was underestimated by the model. The model underestimate of $^7$Be levels in the
warm months is partly due to the sensitivity to spatial sampling in the model, but also suggests
that the mixing of air masses between the PBL and the lower free troposphere is likely too
weak. This suggests that additional work comparing the model results with more surface
observations is needed in order to better understand this effect. The simulated lower annual
average $^7$Be concentration relative to the observation is also partly attributed to the fact that the
model used the $^7$Be production rate for a solar maximum year, while in 2005 (our simulation
year) the solar activity was rather low.

9       By examining the wind fields and horizontal distribution of radiotracers in the model, we

noted that the sampling site is in a location where there is a large gradient, especially in the
North-South direction. Accordingly, we investigated the sensitivity of model results to spatial
sampling. A better agreement between the model and the observations at some adjacent
gridboxes was found. The $^7$Be March maximum was linked to the large stratospheric influence
during winter/spring. The model tends to underestimate the summertime $^{210}$Pb and $^7$Be, but
better simulates the $^7$Be/$^{210}$Pb ratio because the model errors due to precipitation scavenging
appear to be canceled out in the ratio.
We have conducted a series of model sensitivity experiments to further examine and
quantify the roles of wet scavenging, dry deposition, and convection (transport and scavenging)
in controlling the seasonality of $^{210}$Pb and $^7$Be at Mt. Cimone. Dry deposition does not have a
significant effect on the magnitude and seasonality of $^{210}$Pb and $^7$Be concentrations at the site.
The relatively weak combined effects of convective transport and convective scavenging on
the radiotracer seasonality were attributed to the compensating effects of convective transport
and convective scavenging on tracer concentrations in the lower free troposphere (at the
elevation of Mt. Cimone). Convection appears to be more important to the regional distribution
of both radiotracers during summer and autumn, although it is also significant during spring.



Finally, scavenging is found to be the most important process controlling the seasonal
variations of $^{210}$Pb and $^7$Be at Mt. Cimone. For $^{210}$Pb, precipitation plays a more important role
during August-December than during January-July. This was attributed to the seasonality of
local and regional precipitation, which shows prolonged elevated values in the period of
August-December.
While our simulations demonstrated some capabilities of the model to reproduce the
seasonality of $^{210}$Pb and $^7$Be, they highlight the weaknesses of the model in reproducing local
features, presumably due to its coarse resolution. Model simulations at a higher resolution
would improve this model analysis of $^{210}$Pb and $^7$Be observations at Mt. Cimone, a high-
elevation site. The understanding of downward transport associated with convection during
summer also requires improving. As such, $^{210}$Pb and $^7$Be tracers will prove to be very useful in
our understanding of seasonal behaviors of other environmentally important trace gases and
aerosols at Mt. Cimone. Since other aerosols and trace gases (e.g., black carbon, CO, $O_3$) are
also measured at the station, we plan to conduct comparisons between model simulations and
those measurements to corroborate or contrast with the radionuclide results.
**Data availability**
A description of the observational data and model output used in this paper can be found in
Sect. 2 and they are available upon request by contacting Laura Tositti (laura.tositti@unibo.it)
and Hongyu Liu (hongyu.liu-1@nasa.gov), respectively.
**Acknowledgements.** Italian Air Force Meteorological Office (IAFMS) and ISAC-CNR are
gratefully acknowledged for their precious technical support at the Mt. Cimone station. In
particular, ISAC-CNR is gratefully acknowledged for providing infrastructural access at the
WMO-GAW Global Station Italian Climate Observatory "O. Vittori" at Mt. Cimone. IAFMS





is gratefully acknowledged for providing meteorological observations at Mt. Cimone station.
The Italian Climate Observatory "O. Vittori" is supported by MIUR and DTA-CNR throughout
the Project of National Interest NextData. Erika Brattich thanks the National Institute of
Aerospace (NIA) Visitor Program for hosting her one month visit, and the Department of
Biological, Geological and Earth Sciences of the University of Bologna for grant support
during her PhD study. Hongyu Liu is supported by NASA Modeling and Analysis Program
(MAP), NASA Atmospheric Composition Modeling and Analysis Program (ACMAP), and
NASA Atmospheric Composition Campaign Data Analysis and Modeling (ACCDAM)
program. The GMI activity is managed by José Rodriguez and Susan Strahan (NASA GSFC).
Stephen Steenrod, Megan Damon, and Jules Kouatchou (GSFC) are acknowledged for
programming support. NASA Center for Computational Sciences (NCCS) provided
supercomputing resources.

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



3  **Figures**

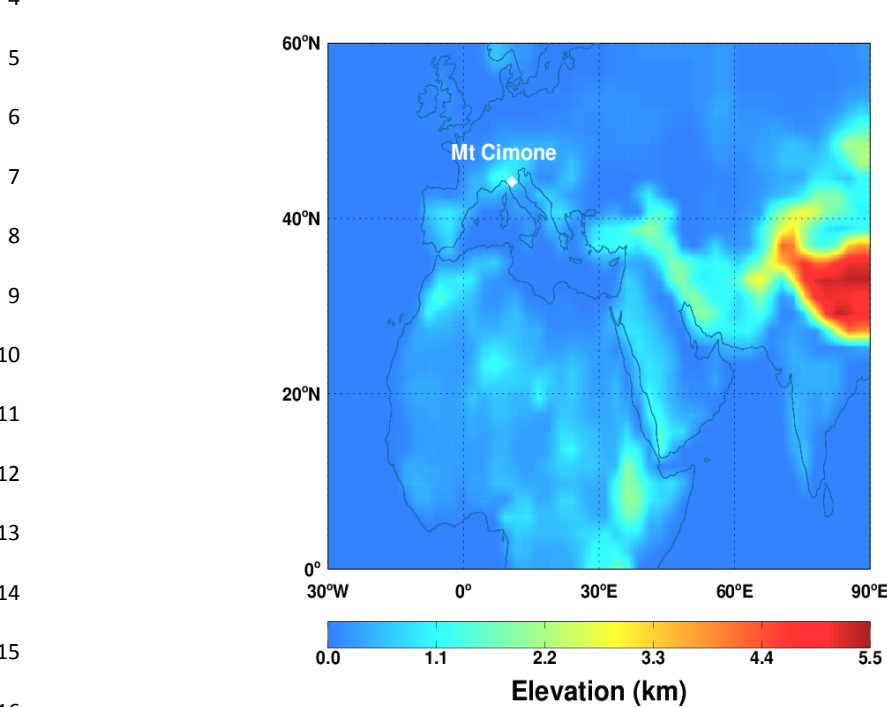

17  **Figure 1.** Surface elevations (km) in the model. The white dot indicates the location of Mt.

18  Cimone (44°12' N, 10°42' E, 2165 m asl).





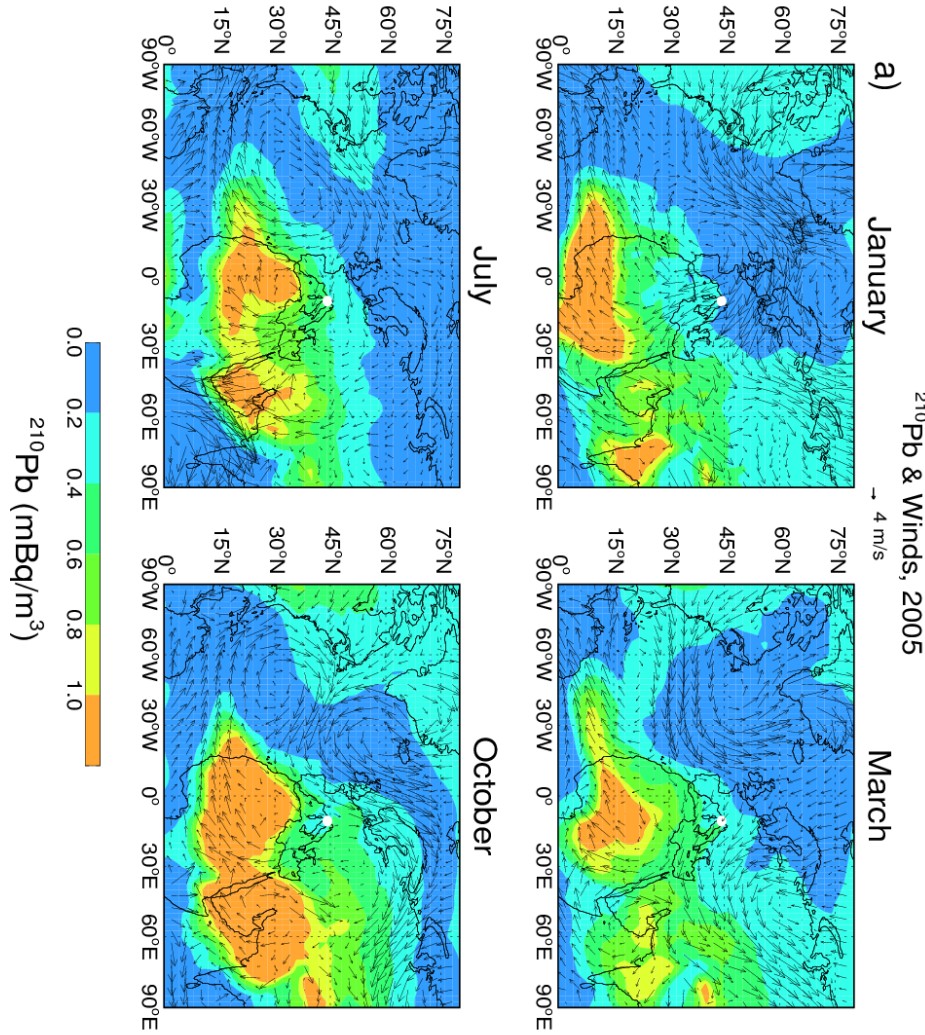

**Figure 2.** Simulated monthly mean (a) [210]Pb concentrations and (b) [7]Be concentrations, at the
elevation of Mt. Cimone. Arrows represent the seasonality of winds in the MERRA
meteorological data. The white dot indicates the location of Mt. Cimone (44°12' N, 10°42' E,
2165 m asl). To be continued.





2 **Figure 2.** (continued)





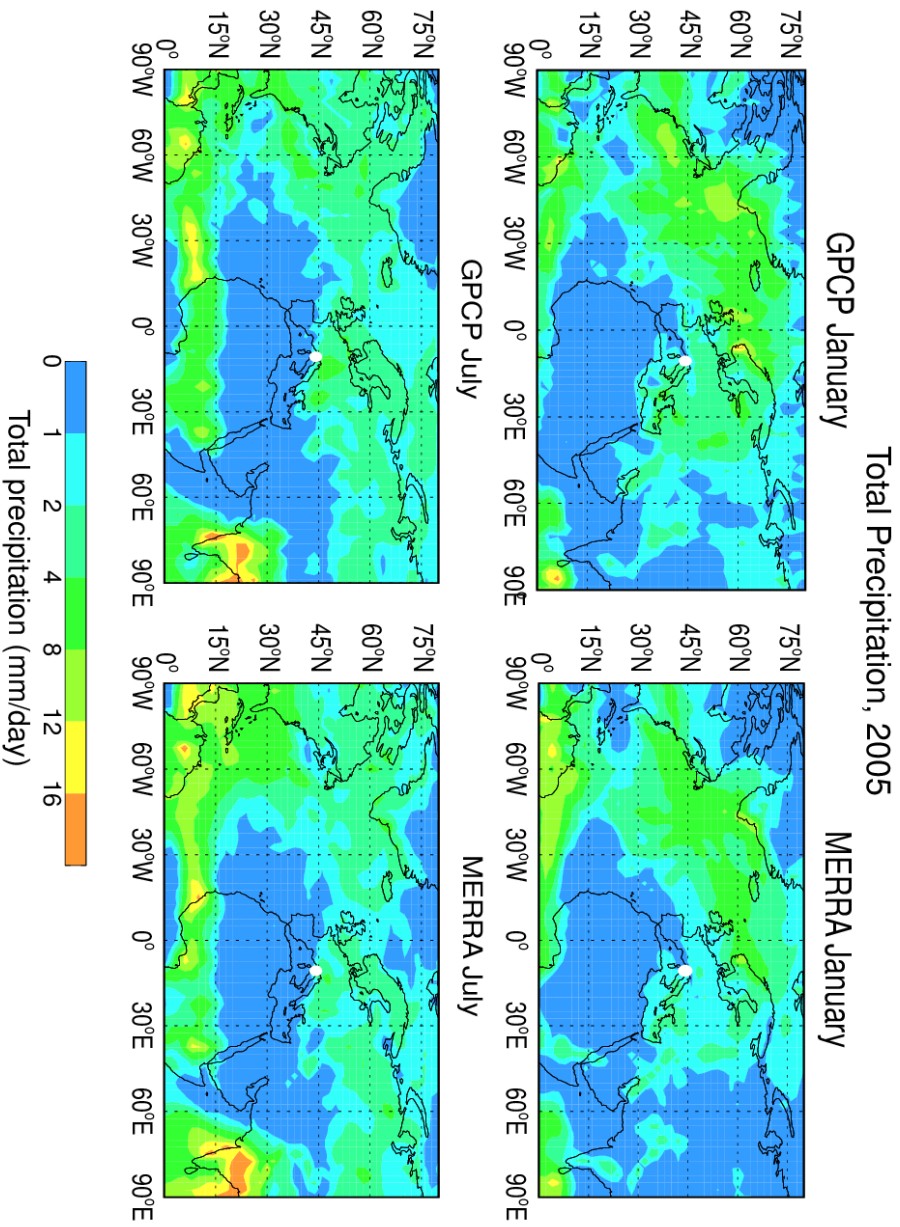

**Figure 3.** Comparison of the MERRA total precipitation (0-75°N, 90°W-90°E) during January

and July 2005 with that in the GPCP observations. The white dot indicates the location of Mt.

Cimone (44°12'N, 10°42'E, 2165 m asl).





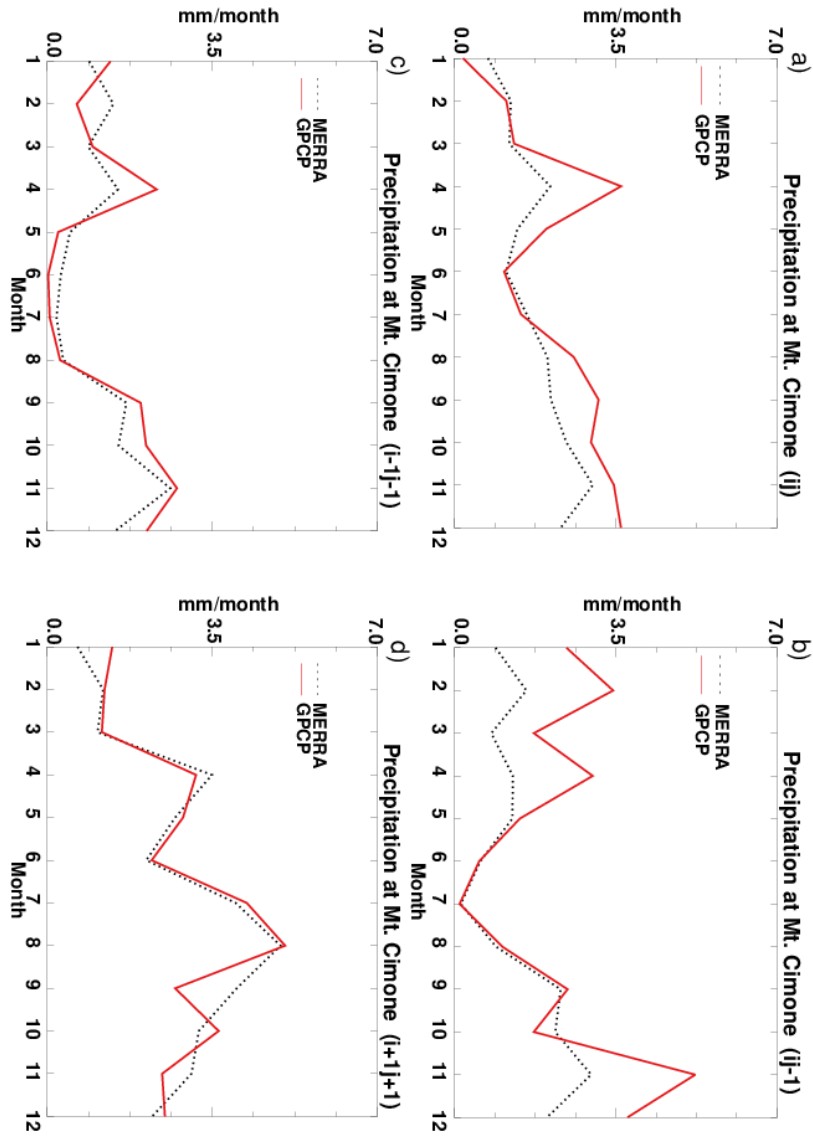

**Figure 4.** Comparison of the seasonal precipitation at Mt. Cimone in the MERRA

meteorological data set with that in the GPCP observations for (a) the model gridbox ("ij")

corresponding to the location of Mt. Cimone, (b) the model gridbox ("ij-1") to the west of "ij",

(c) the model gridbox ("i-1j-1") to the southwest of "ij", and (d) the model gridbox ("i+1j+1")

to the northeast of "ij".





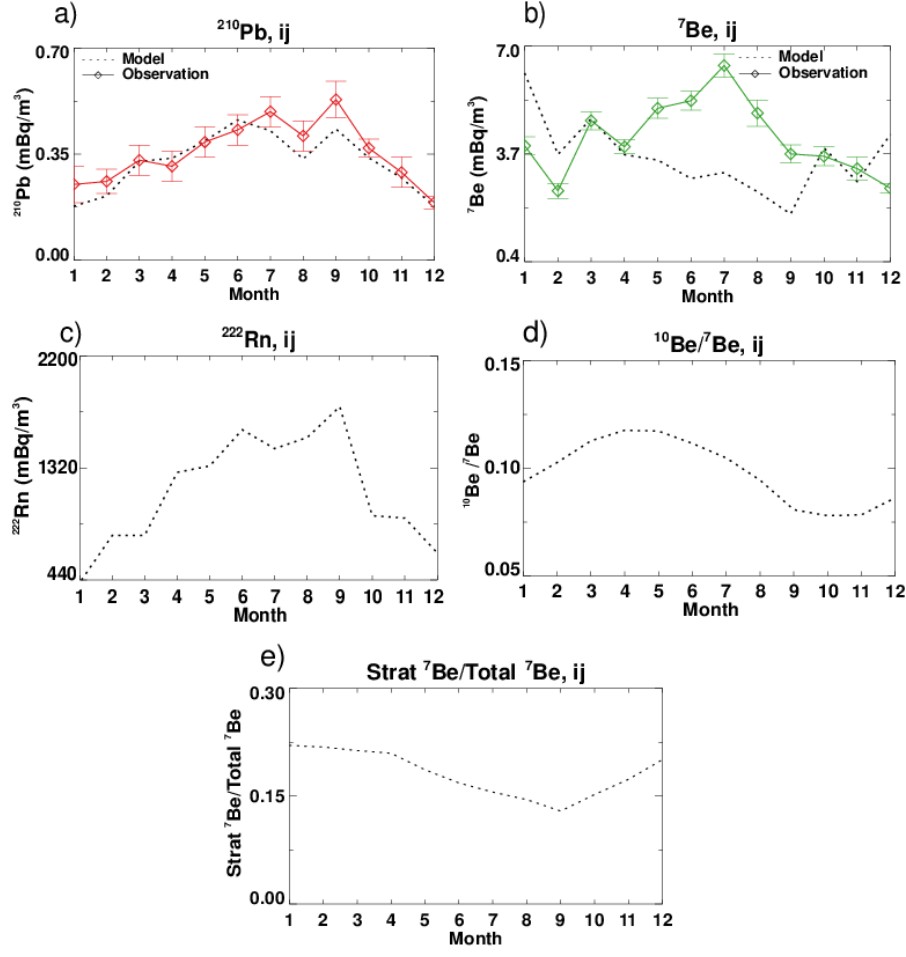

**Figure 5 (a,b,c,d,e).** Comparison of GMI simulated (black dotted line) monthly (a) $^{210}$Pb and

(b) $^{7}$Be activities with those observed at Mt. Cimone (solid lines) in 2005. Also shown are GMI

simulated monthly activities of (c) $^{222}$Rn, (d) $^{10}$Be/$^{7}$Be ratios, and (e) strat $^{7}$Be/total $^{7}$Be ratios.

Model values are for the "ij" gridbox corresponding to the location of Mt. Cimone. Vertical

bars indicate the uncertainty in observed activities.





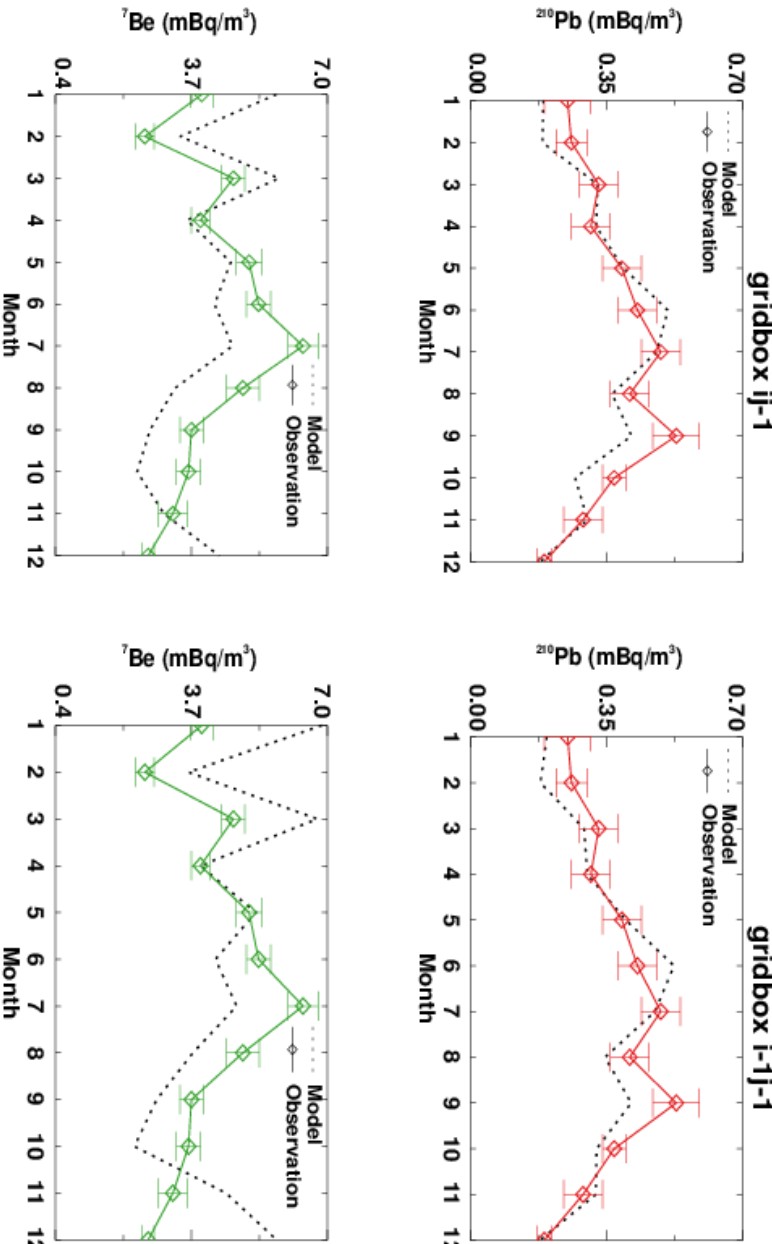

2      **Figure 6.** Same as Figure 5(ab), but for the "ij-1" to the south of "ij" (left column) and "i-1j-

3      1" to the southwest of "ij" (right column) grids, respectively.



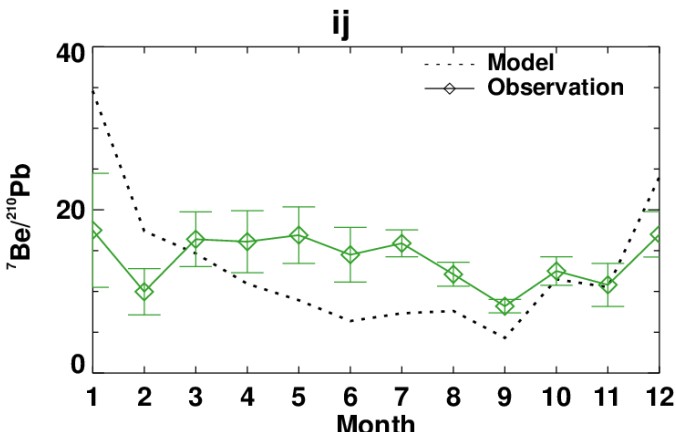

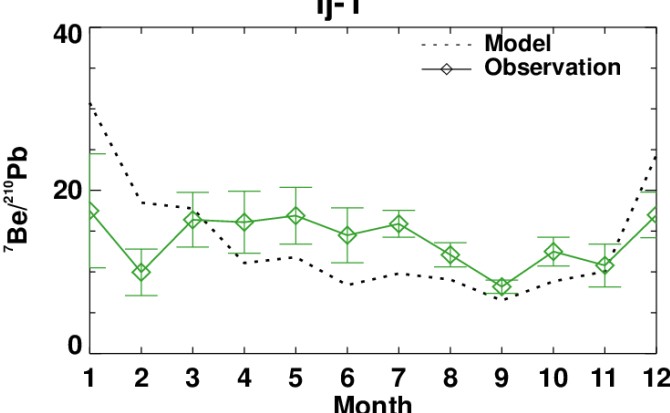

**Figure 7.** Comparison between GMI simulated monthly $^{7}$Be/$^{210}$Pb ratios at the "ij" and "ij-1"

grids (black dotted line) and those from the observations at Mt. Cimone (green solid line).

Vertical bars indicate the uncertainty in observed activities.





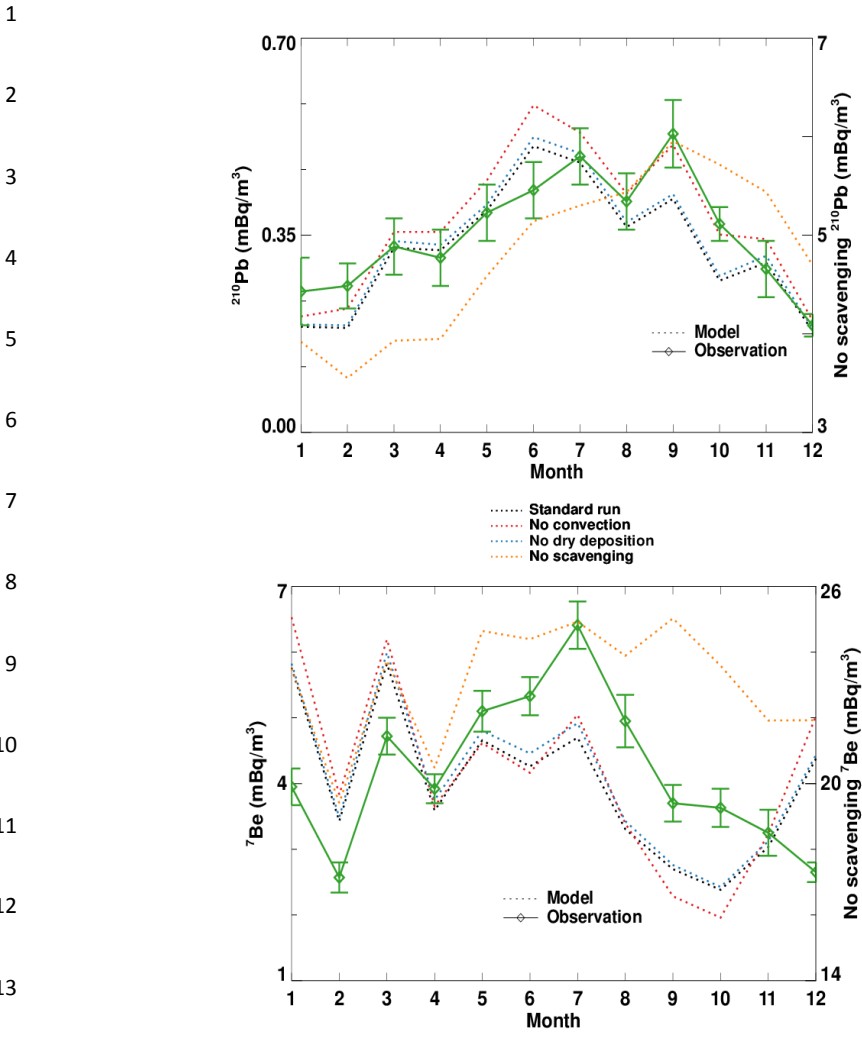

**Figure 8.** Comparison of GMI simulated monthly $^{210}$Pb and $^{7}$Be activities at Mt. Cimone between the standard (black dotted line) and the sensitivity runs ("ij-1" grid). The sensitivity runs are those without convective transport/scavenging (red dotted line), without dry deposition (blue dotted line), and without scavenging (orange dotted line; y-axis on the right). The observations are shown as green solid line. Vertical bars indicate the uncertainty in observed activities.





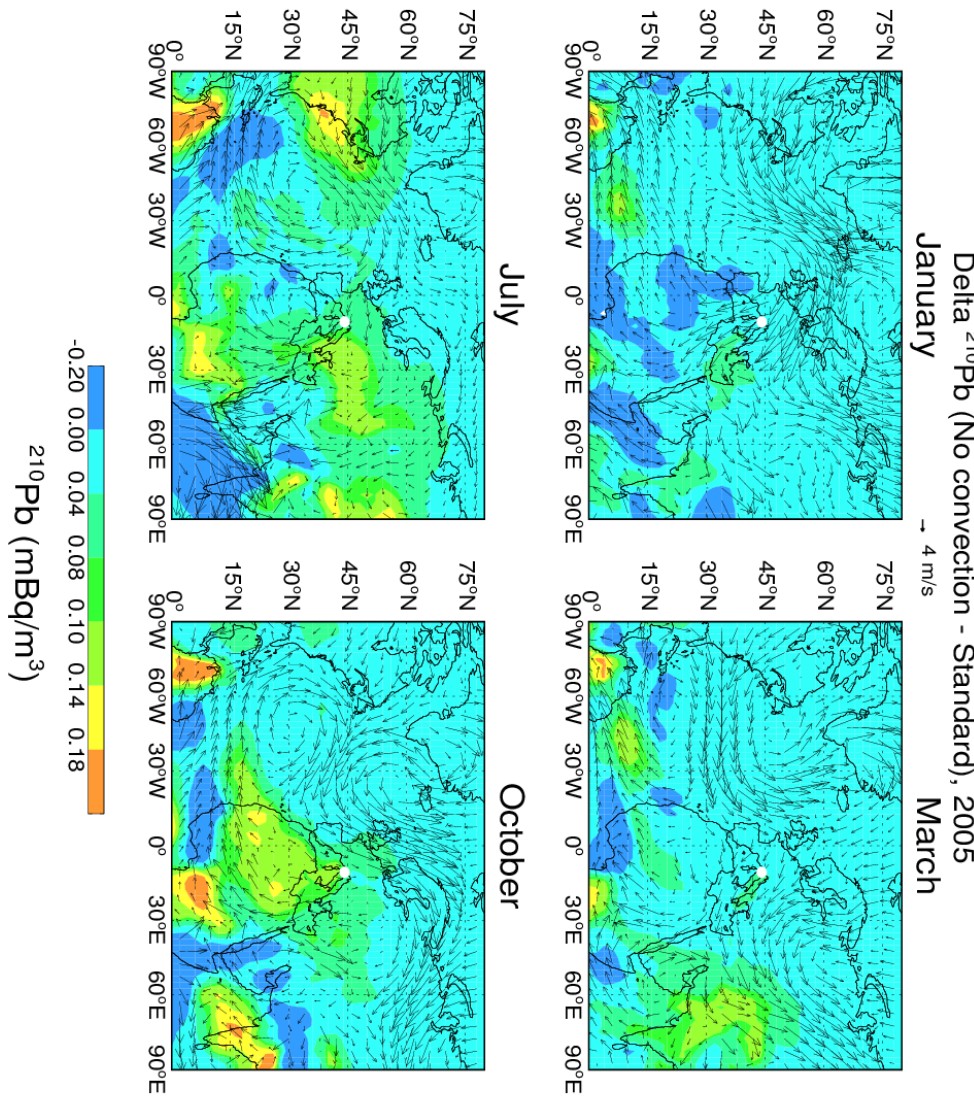

**Figure 9.** GMI simulated differences of $^{210}$Pb concentrations at the elevation of Mt. Cimone

between a sensitivity run without convection (i.e., without transport and scavenging in

convective updrafts) and the standard run. Arrows denote MERRA winds. The white dot

indicates the location of Mt. Cimone (44°12' N, 10°42' E, 2165 m asl).



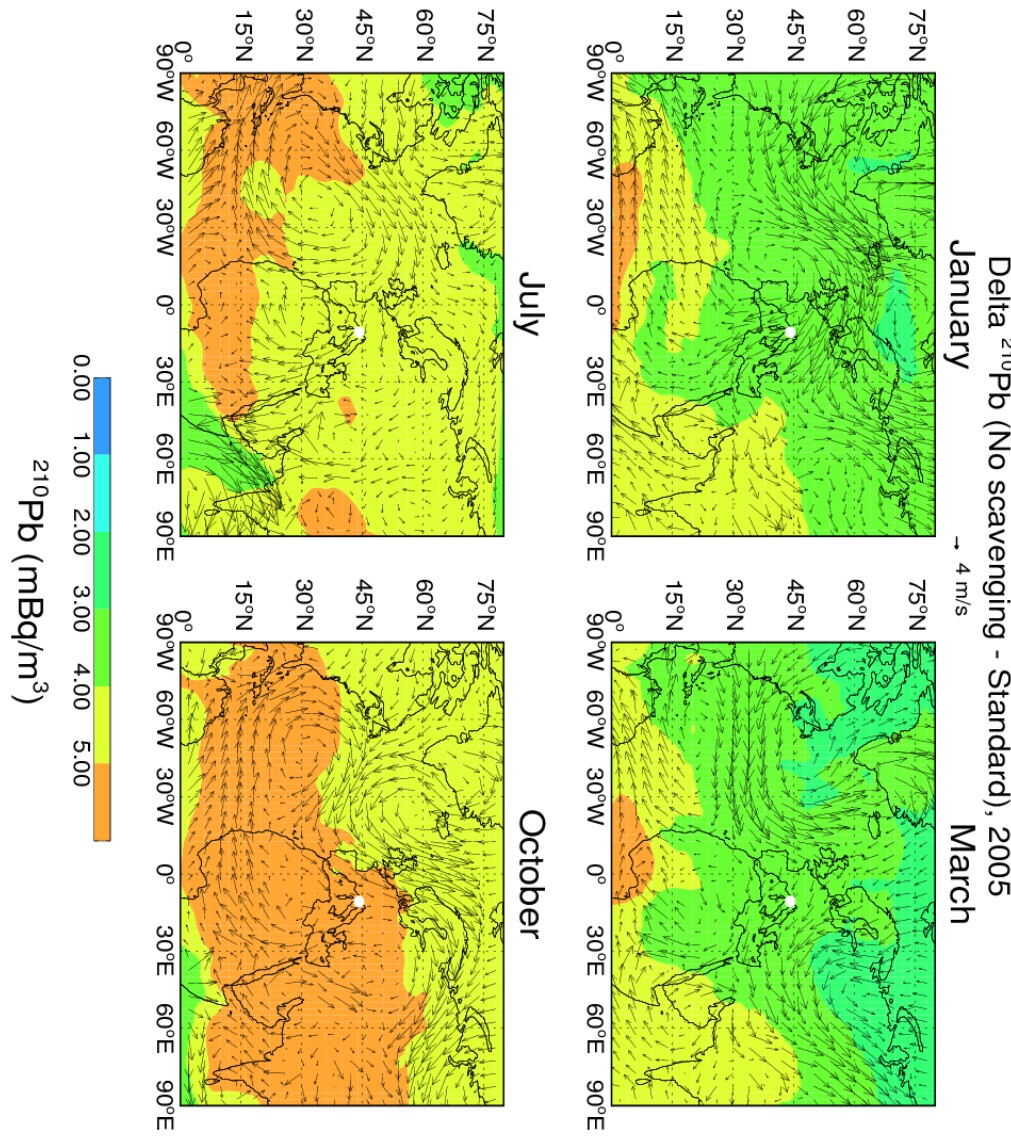

**Figure 10.** Same as Figure 9, but for a sensitivity simulation where wet scavenging is turned

off.





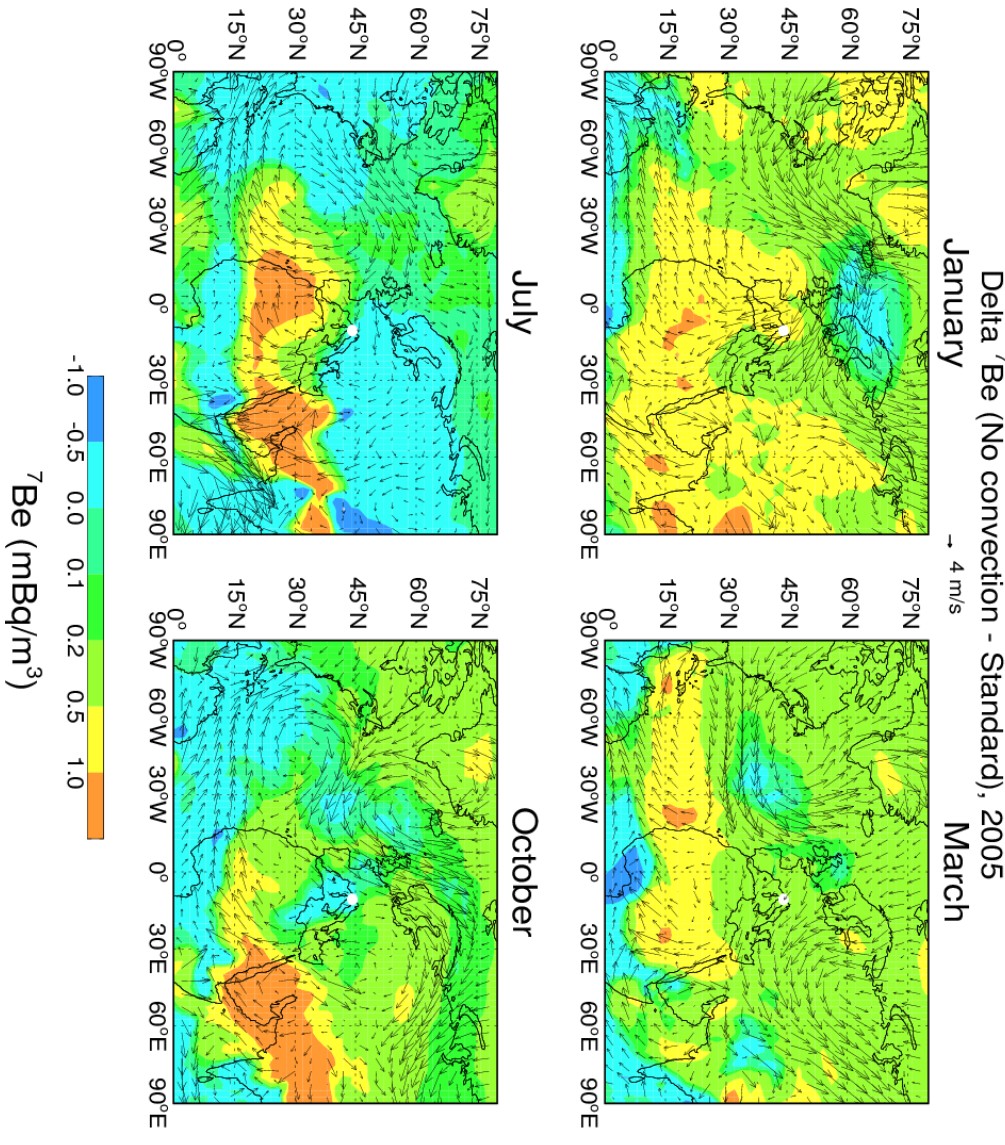

2   **Figure 11.** GMI simulated differences of [7]Be concentrations at the elevation of Mt. Cimone

3   between a sensitivity run without convection and the standard run. Arrows denote MERRA

4   winds. The white dot indicates the location of Mt. Cimone (44°12' N, 10°42' E, 2165 m asl).





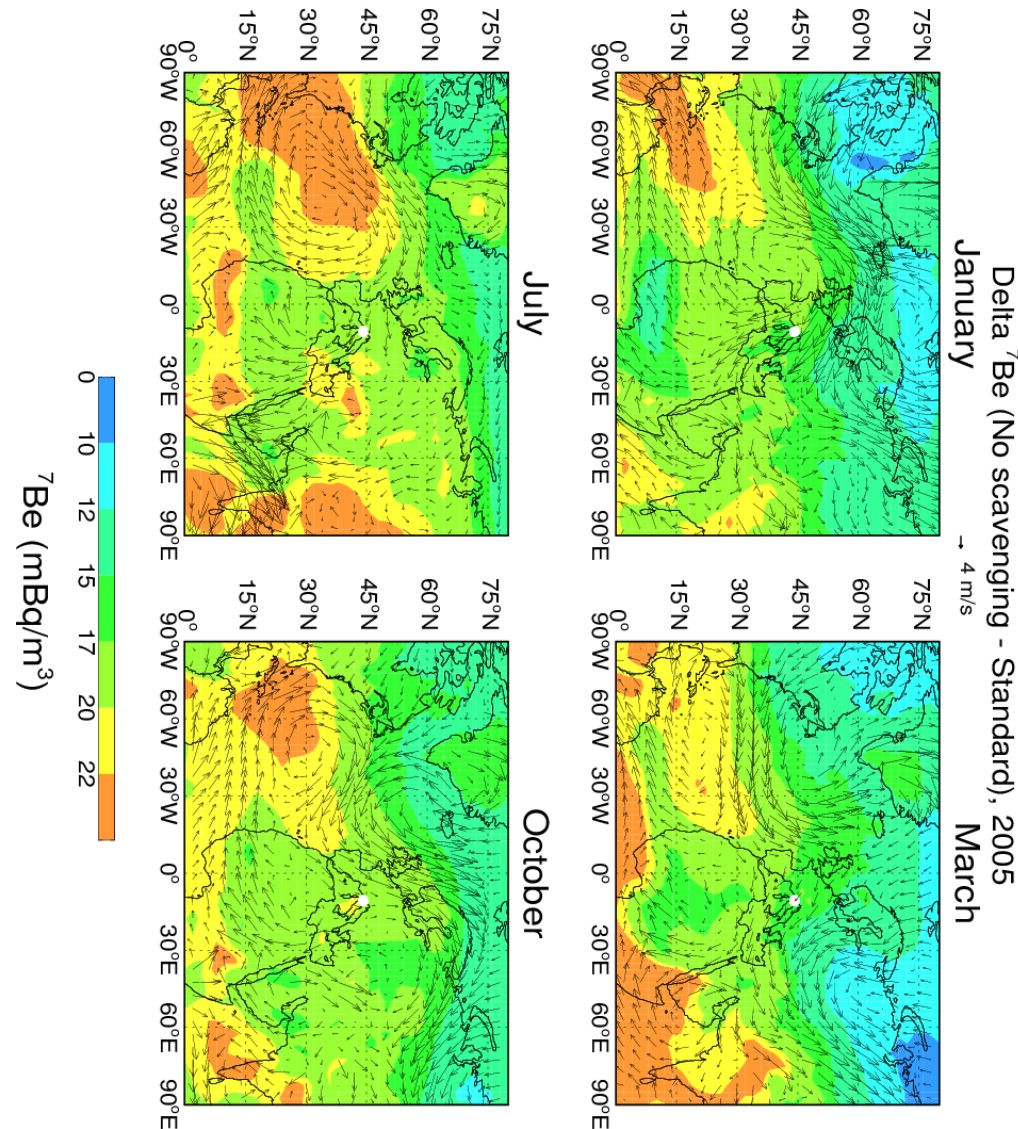

3 **Figure 12.** Same as Figure 11 but for the difference between a sensitivity run without wet

4 scavenging and the standard run.

