# Peer review of "Processes controlling the seasonal variations of 210Pb and 7Be"

_Atmospheric Chemistry and Physics, 2016_

## Referee Comment (RC1) · Anonymous Referee #1 · 30 Aug 2016

The submitted manuscript aims to contribute in the understanding of the roles of transport and precipitation scavenging processes in controlling seasonality of Pb-210 and Be-7 radionuclide aerosol tracers at Mt. Cimone, Italy using a global chemistry transport model. It has an added value and I suggest acceptance of the manuscript for publication after taking into consideration the following comments.

Comments 1) page 10, line 9: What is the spatial resolution of the model simulations? 2) page 11, lines 3-6,: The authors state " For the simulations of radionuclides, each simulation was run for six years, recycling the meteorological data for each year of the simulation, to equilibrate the lower stratosphere as well as the troposphere". Does this practically mean that it is simulated the same year for six times and that the first five

years were used as a spin up time? Also please mention again here that the actual year of the simulation is 2005. 3) page 13, lines 20-21: The authors state that " In the model Mt. Cimone appears to be in a location where there is a large horizontal gradient of wind (transport)." Mind though that the model's winds in Figure 2 are from specific months in a single year (the year 2005) and hence do not actually represent a wind climatology of the respective months. 4) page 14, line 10-14: Note also that the etesian wind system at eastern Mediterranean in July is also well represented in Figure 2. 5) page 15, line 11-14: The authors state that "Large differences between the MERRA precipitation and that locally observed at the station are instead present (not shown): in particular, the MERRA precipitation is larger during winter-autumn, while it is much more similar to that observed during spring-summer." I would suggest to add information or a graph with the station-based observations of precipitation at Mt Cimone (even as supplementary material). Of course, MERRA data reflect large scale precipitation features while the station-based observations reflect local features. Nevertheless in your analysis you compare modelled Pb-210 and Be-7 radionuclide concentrations with the respective station based measurements at Mt Cimone, but these station based radionuclide measurements are presumably linked more with the local observation of precipitation than with large scale MERRA precipitation data. 6) page 17, line 21-23: The authors state that "The correlation between observed and simulated monthly 7Be activities also increases from $R^2 = 0.03$ at "ij" to $R^2 = 0.11-0.60$ at adjacent model gridboxes." Please specify at which grid-box you get 0.6 and discuss the reason for this considerable improvement. 7) page 17, line 21-23: The authors state that " As for 7Be, the model well captures the March maximum (i.e., secondary maximum in the observations) and the general seasonal pattern during the cold and transition seasons." I think that this statement is not very consistent with Figure 5b. Actually, according to Figure 5b the model does not seem to capture the general seasonal pattern for Be-7. 8) page 18, line 19-21: The authors state that " The simulated seasonal pattern of the 10Be/7Be ratio is very similar to the observations at Zugspitze (Germany, 2962 m asl) (Zanis et al., 2003), characterized by a not-pronounced seasonal cycle" . In fact

the simulated Be-10/Be-7 ratio in Figure 5d has a clear seasonal cycle and looking the respective graph Figure 3 from the cited paper of Zanis et al., 2003, I see a better agreement with Jungfraujoch than with Zugspitze. 9) page 19, line 9-11: The authors state that " However, the model tends to overestimate the observed 7Be concentrations and 7Be/210Pb ratios during December-February, suggesting that STE and/or subsidence in the model is likely too fast in this region." This is a rather speculative comment. It needs more justification. What do you mean with too fast? Maybe stronger STE fluxes? Are there any references showing how the STE fluxes of this model compares with other global CTMS or GCMs? 10) page 19, line 11-13: The authors state that "As reported by Huang et al. (2013), a stronger net subsidence of air masses to the surface could be due to unrealistic meteorological conditions (e.g., boundary layer structure, wind fields, vertical mixing)." This is a rather general comment. Is this true for the meteorological data used here in the CTM? Please clarify this issue.

11) page 20, line 19-20: The authors state that "The model annual average biases are about 8% for 210Pb and about 19% for 7Be, respectively. By contrast, the model average bias for 7Be/210Pb ratios is about -13% (Figure 7)." Please comment on the error propagation on the ratio. 12) page 22, line 8-9: The authors state that " ... suggesting that large-scale circulation in this region with complex topography may not be resolved by the coarse-resolution model." I guess you mean that regional and local circulations are not resolved by the global model. 13) page 24, line 1-4: The authors state that " The model underestimate of 7Be levels in the warm months is partly due to the sensitivity to spatial sampling in the model, but also suggests that the mixing of air masses between the PBL and the lower free troposphere is likely too weak." If the model mixing between the PBL and the lower free troposphere becomes stronger then this will result in more mixing of PBL air poor in Be-7 with free tropospheric air, hence even smaller concentrations of Be-7 and larger model underestimate of Be-7 at Mt Cimone.

14) To my understanding, the authors claim that the CTM cannot capture the observed seasonal cycle of Be-7 with a summer max at Mt Cimone because of local features

which are not resolved in the model. However mind that the summer maximum Be-7 at Mt Cimone is also apparent at Jungfraujoch, Sonnblick and Zugspitze (see e.g. Figure 7 in Gerasopoulos et al., 2001). So maybe this feature does not seem to be a very local phenomenon but is rather of larger horizontal scale.
* * *

---

## Referee Comment (RC2) · Anonymous Referee #2 · 10 Oct 2016

This manuscript uses the Global Modeling Initiative chemistry and transport model (GMI-CTM) driven by the NASA's MERRA assimilated data to simulate the seasonal variations of 210Pb and 7Be at the WMO-GAW station of Mt. Cimone (44°12' N, 10°42' E, 2165 m asl, Italy). The GMI-CTM model is well evaluated by using the observed seasonal variations of 210Pb and 7Be at Mt. Cimone in this study. The model well reproduced the seasonal pattern of 210Pb concentrations at the site together with very reasonable mechanisms controlling the seasonal 210Pb variation. However, the model failed in reproducing the seasonal pattern of 7Be, particularly the underestimations of 7Be in the summer season. The authors performed model sensitivity experiments and found that the wet scavenging process designed in the model is the dominant reason

for 7Be underestimations, and the next is the low sensitivity caused by the coarse spatial resolution of model. The manuscript is interesting with important significance and high quality of science, and presentation is good. I have the following suggestions for the authors to consider.

1)How about the actual precipitation at Mt. Cimone in Figure 4? The difference between actual, modeled, and GPCP precipitations would support that wet scavenging is the main reason controlling 7Be seasonal variations shown in Figure 8. Please notice the precipitation comparisons in Figure 4 (ij), which shows that the model precipitation is generally lower than that of GPCP, meaning that the modeled wet scavenging processes perhaps is lower than the reality. This weak modeled wet scavenging seems to be very significant for the 7Be concentrations shown in Figure 8.

2)One section can be added to illustrate the model results of this study in comparison with historical model studies. Those model studies may include as follows: 7Be: Brost, et al., J Geophys Res, 96, 1991; 210Pb: Feichter,et al., J. Geophys. Res., 96, 1991; Lee, H. N., et al., J Geophys. Res., 109, D22203, 2004, doi: 10.1029/2004JD005061. 7Be/210Pb: Koch et al., J Geophys. Res, 101(D13), 1996.

3)The WMO-GAW station, Mt. Cimone (44°12' N, 10°42' E, 2165 m asl, Italy) is quite close to the Apls stations, such as Jungfraujoch (46.32°N, 7.59°E, elevation 3580 m asl) and Zugspitze(47. °N, 11.0°E, 2962 m a.s.l.) in the model grids. How about the general results of the model and observation comparisons for those two stations in 2005? I believe these comparisons will support the conclusion that coarse of the model runs is one of the reasons for the worse 7Be comparisons.

4)For Figure 8, I am confused that without the wet-scavenging process, the 210Pb concentration is even lower than that of observed from January to July. The convection uplift of 222Rn seems does not support the summer 210Pb maximum but on the contradictory. How about the sensitivity experiments with case of 7Be/210Pb in Figure 8? Why do you show the sensitivity test for ji-1 grid rather than the ji grid in this figure?

---

## Editor Comment (EC1) · J. Ma (Editor) · 11 Oct 2016

The authors may consider giving a brief description of what the physical and chemical processes are included in their model for the simulation of radionuclides Pb-210 and Be-7 in both the stratosphere and the troposphere, instead of just referring to the literatures (Page 10, Line 9-14).

Pb-210 and Be-7 were analyzed from the PM10 samples according to a description of the radionuclide measurement (Page 9, Line 5-8). I wonder if the treatment of aerosol processes in the model could affect the simulation of radionuclides. How does the model treat the uptake of radionuclides on particulate matters? Does the uncertainties in the simulation of PM10 affect the simulation results of Pb-210 and, in particular,

[Figure]

Be-7?

The treatment of Be-7 source in the model is also suggested to be described more specifically, e.g., by providing the prescribed concentrations or by giving the production rates at certain altitudes.

---

## Author Comment (AC1) · 28 Nov 2016

We thank Reviewer # 1 for the detailed comments on our manuscript. Below please find our itemized replies.

1) page 10, line 9: What is the spatial resolution of the model simulations?

Reply – We have added this information in the text: "Meteorological data used to drive the CTM at 2° latitude by 2.5° longitude resolution, e.g., horizontal winds, convective mass fluxes and precipitation fields, are the Modern-Era Retrospective analysis for Research and Applications (MERRA) assimilated data set from the NASA Global Modeling and Assimilation Office (GMAO) (Rienecker et al., 2011)."

[Figure]

2) page 11, lines 3-6: The authors state " For the simulations of radionuclides, each simulation was run for six years, recycling the meteorological data for each year of the simulation, to equilibrate the lower stratosphere as well as the troposphere". Does this practically mean that it is simulated the same year for six times and that the first five years were used as a spin up time? Also please mention again here that the actual year of the simulation is 2005.

Reply – Yes. Thanks for pointing it out. Now the text reads "For the simulations of radionuclides, each simulation was run for six years, recycling the MERRA meteorological data for 2005, to equilibrate the lower stratosphere as well as the troposphere (Liu et al., 2001). The sixth-year output was used for analysis."

3) page 13, lines 20-21: The authors state that " In the model Mt. Cimone appears to be in a location where there is a large horizontal gradient of wind (transport)." Mind though that the model's winds in Figure 2 are from specific months in a single year (the year 2005) and hence do not actually represent a wind climatology of the respective months.

Reply – Indeed, the model's winds in Figure 2 are from specific months in a single year (2005) and do not represent a wind climatology of the respective months. However, we do not mean to represent a wind climatology here. We have revised the sentence to "In the model Mt. Cimone appears to be in a location where there is a large horizontal gradient of wind (transport) during 2005."

4) page 14, line 10-14: Note also that the etesian wind system at eastern Mediterranean in July is also well represented in Figure 2.

Reply – Indeed. We have added a sentence at the end of this paragraph: "However, MERRA is able to capture the summertime north-north easterly winds in the eastern Mediterranean (Aegean Sea), known as the Etesian winds, generated by thermal effects."

5) page 15, line 11-14: The authors state that "Large differences between the MERRA precipitation and that locally observed at the station are instead present (not shown): in particular, the MERRA precipitation is larger during winter-autumn, while it is much more similar to that observed during spring-summer." I would suggest to add information or a graph with the station-based observations of precipitation at Mt Cimone (even as supplementary material). Of course, MERRA data reflect large scale precipitation features while the station-based observations reflect local features. Nevertheless in your analysis you compare modelled Pb-210 and Be-7 radionuclide concentrations with the respective station based measurements at Mt Cimone, but these station based radionuclide measurements are presumably linked more with the local observation of precipitation than with large scale MERRA precipitation data.

Reply - We thank the reviewer for the suggestion. As reported later on in the manuscript (page 15, lines 16-22) the local precipitation pattern at Mt. Cimone is different from the regional pattern of the surrounding area, and this difference could partially explain the disagreement between the observed and simulated pattern of precipitation. As commented by the reviewer and discussed in the paper (page 14, lines 23-25; page 15, lines 1-2), local precipitation at the site is important to the scavenging of radionuclides and the difference between the observed and MERRA precipitation could contribute to the biases in our model simulations due to the errors in the precipitation scavenging of radionuclides. We have added information and revised the text to "Large differences between the MERRA precipitation and that locally observed at the station are instead present. While the daily mean observed 2005 precipitation is 0.81 mm, which is close to the corresponding precipitation (0.73 mm) in MERRA at the "ij" grid (i.e., a negative bias of -0.08 mm); the model bias is positive and much higher (0.31 – 1.28 mm) at adjacent grids. This bias may very well reflect again the fact that the observed surface precipitation is localized, whereas the satellite and MERRA precipitations correspond to a much larger scale (about 200 km)."

6) page 17, line 21-23: The authors state that "The correlation between observed and

simulated monthly 7Be activities also increases from R2 = 0.03 at "ij" to R2 = 0.11-0.60 at adjacent model gridboxes." Please specify at which grid-box you get 0.6 and discuss the reason for this considerable improvement.

Reply – The revised text reads "The largest value of R2 = 0.6 was obtained at the "ij-1" gridbox to the south of "ij" (Figure 6). This improvement is due to the large horizontal gradient in the simulated 7Be concentrations near the site (Figure 2)."

7) page 17, line 21-23: The authors state that " As for 7Be, the model well captures the March maximum (i.e., secondary maximum in the observations) and the general seasonal pattern during the cold and transition seasons." I think that this statement is not very consistent with Figure 5b. Actually, according to Figure 5b the model does not seem to capture the general seasonal pattern for Be-7.

Reply – To avoid confusion, we have revised the sentence to "As for 7Be, the model well captures the March maximum (i.e., secondary maximum in the observations) and the month-to-month variation during the cold and transition seasons (January-April, October-December)."

8) page 18, line 19-21: The authors state that " The simulated seasonal pattern of the 10Be/7Be ratio is very similar to the observations at Zugspitze (Germany, 2962 m asl) (Zanis et al., 2003), characterized by a not-pronounced seasonal cycle". In fact the simulated Be-10/Be-7 ratio in Figure 5d has a clear seasonal cycle and looking the respective graph Figure 3 from the cited paper of Zanis et al., 2003, I see a better agreement with Jungfraujoch than with Zugspitze.

Reply - Thanks the reviewer for pointing this out to us. Accordingly, we have revised the text to "The simulated seasonal pattern of the 10Be/7Be ratio is very similar to the observations at Jungfraujoch (Switzerland, 3580 m asl) (Zanis et al., 2003), characterized by a clear seasonal cycle with peak ratios in spring."

9) page 19, line 9-11: The authors state that " However, the model tends to overestimate the observed 7Be concentrations and 7Be/210Pb ratios during December-February, suggesting that STE and/or subsidence in the model is likely too fast in this region." This is a rather speculative comment. It needs more justification. What do you mean with too fast? Maybe stronger STE fluxes? Are there any references showing how the STE fluxes of this model compares with other global CTMS or GCMs?

Reply – This statement is for the site of Mt. Cimone and year 2005, and is only suggestive. To address the reviewer's concern, we have added a new reference and revised the text to "However, the model tends to overestimate the observed 7Be concentrations and 7Be/210Pb ratios during December-February, suggesting that stratospheric influence and/or subsidence in the model is probably too strong in this region at this time of the year. It is noted that globally integrated STT mass fluxes in the MERRA reanalysis are actually smaller than in some other reanalyses, e.g., ERA-Interim, JRA-55, and MERRA-2 (Boothe and Homeyer, 2016)."

10) page 19, line 11-13: The authors state that "As reported by Huang et al. (2013), a stronger net subsidence of air masses to the surface could be due to unrealistic meteorological conditions (e.g., boundary layer structure, wind fields, vertical mixing)." This is a rather general comment. Is this true for the meteorological data used here in the CTM? Please clarify this issue.

Reply – To avoid confusion, we have removed this sentence.

11) page 20, line 19-20: The authors state that "The model annual average biases are about 8% for 210Pb and about 19% for 7Be, respectively. By contrast, the model average bias for 7Be/210Pb ratios is about -13% (Figure 7)." Please comment on the error propagation on the ratio.

Reply – We comment on the error propagation on the ratio after this statement: "The smaller model bias for 7Be/210Pb ratios than for 7Be concentrations reflects the fact that the ratio cancels out the errors in precipitation scavenging (Koch et al. 1996) that contribute to the underestimate of 210Pb and 7Be activities. On the other hand, the

negative model bias for the 7Be/210Pb ratio again points to weak downward mixing from the free troposphere."

12) page 22, line 8-9: The authors state that " ... suggesting that large-scale circulation in this region with complex topography may not be resolved by the coarse-resolution model." I guess you mean that regional and local circulations are not resolved by the global model.

Reply – Indeed. We have revised the sentence to "None of our simulations is able to describe the observed 7Be summertime peak, suggesting that local and regional circulations in this region with complex topography may not be resolved by the coarse-resolution model."

13) page 24, line 1-4: The authors state that "The model underestimate of 7Be levels in the warm months is partly due to the sensitivity to spatial sampling in the model, but also suggests that the mixing of air masses between the PBL and the lower free troposphere is likely too weak." If the model mixing between the PBL and the lower free troposphere becomes stronger then this will result in more mixing of PBL air poor in Be-7 with free tropospheric air, hence even smaller concentrations of Be-7 and larger model underestimate of Be-7 at Mt Cimone.

Reply - The vertical mixing between the PBL and the lower free troposphere includes both an upward motion from the PBL to the lower free troposphere (poor in 7Be), and a downward motion from the lower free troposphere to the PBL (richer in 7Be). We have changed the sentence to "The model underestimate of 7Be levels in the warm months is partly due to the sensitivity to spatial sampling in the model, but also suggests that the mixing of air masses between the PBL and the lower free troposphere (e.g., via convection and compensating subsidence) is likely too weak during summer when the Mt. Cimone station is located within the PBL."

14) To my understanding, the authors claim that the CTM cannot capture the observed seasonal cycle of Be-7 with a summer max at Mt Cimone because of local features

which are not resolved in the model. However mind that the summer maximum Be-7 at Mt Cimone is also apparent at Jungfraujoch, Sonnblick and Zugspitze (see e.g. Figure 7 in Gerasopoulos et al., 2001). So maybe this feature does not seem to be a very local phenomenon but is rather of larger horizontal scale.

Reply - The fact that the CTM cannot capture the observed seasonal cycle of 7Be is due to a combination of factors. Firstly, results show sensitivity to spatial sampling in the model, which can be clearly seen from a better simulated 7Be seasonal cycle at some adjacent gridboxes. Secondly, the summer 7Be maximum observed at mountain sites such as Mt. Cimone, Jungfraujoch, Sonnblick, and Zugspitze results from downward transport of 7Be due to compensating subsidence associated with summertime convective mixing (Gerasopoulos et al., 2001), which the coarse-resolution model may not be able to correctly represent.

---

## Author Comment (AC2) · 28 Nov 2016

We thank reviewer #2 for helpful comments. Below please find our itemized replies.

1) How about the actual precipitation at Mt. Cimone in Figure 4? The difference between actual, modeled, and GPCP precipitations would support that wet scavenging is the main reason controlling 7Be seasonal variations shown in Figure 8. Please notice the precipitation comparisons in Figure 4 (ij), which shows that the model precipitation is generally lower than that of GPCP, meaning that the modeled wet scavenging processes perhaps is lower than the reality. This weak modeled wet scavenging seems to be very significant for the 7Be concentrations shown in Figure 8.

[Figure]

Reply – We thank the reviewer for pointing this out. We have revised the text as follows: "The MERRA precipitation is generally lower than that of GPCP at two gridboxes (except for summer, Figure 4ab), but in good agreement at the other two gridboxes (Figure 4cd). Large differences between the MERRA precipitation and that locally observed at the station are instead present. While the daily mean observed 2005 precipitation is 0.81 mm, which is close to the corresponding precipitation (0.73 mm) in MERRA at the "ij" grid (i.e., a negative bias of -0.08 mm); the model bias is positive and much higher (0.31 – 1.28 mm) at adjacent grids. This bias may very well reflect again the fact that the observed surface precipitation is localized, whereas the satellite and MERRA precipitations correspond to a much larger scale (about 200 km)."

2) One section can be added to illustrate the model results of this study in comparison with historical model studies. Those model studies may include as follows: 7Be: Brost, et al., J Geophys Res, 96, 1991; 210Pb: Feichter,et al., J. Geophys. Res., 96, 1991; Lee, H. N., et al., J Geophys. Res., 109, D22203, 2004, doi: 10.1029/2004JD005061. 7Be/210Pb: Koch et al., J Geophys. Res, 101(D13), 1996.

Reply – Thanks for the suggestion. Since our focus is on the model analysis of observational data from a single station (versus global simulations of 210Pb, 7Be, and 7Be/210Pb), we have decided to cite these historical model studies in various places of the text.

3) The WMO-GAW station, Mt. Cimone (44°12' N, 10°42' E, 2165 m asl, Italy) is quite close to the Alps stations, such as Jungfraujoch (46.32°N, 7.59°E, elevation 3580 m asl) and Zugspitze (47. °N, 11.0°E, 2962 m a.s.l.) in the model grids. How about the general results of the model and observation comparisons for those two stations in 2005? I believe these comparisons will support the conclusion that coarse of the model runs is one of the reasons for the worse 7Be comparisons.

Reply – Unfortunately, we cannot compare the results of our simulations with the observations from Jungfraujoch and Zugspitze stations in 2005. We own only the Mt.

Cimone data, and the observational data from other stations are not publicly available.

4) For Figure 8, I am confused that without the wet-scavenging process, the 210Pb concentration is even lower than that of observed from January to July. The convection uplift of 222Rn seems does not support the summer 210Pb maximum but on the contradictory. How about the sensitivity experiments with case of 7Be/210Pb in Figure 8? Why do you show the sensitivity test for ji-1 grid rather than the ji grid in this figure?

Reply - The model result without scavenging is not lower than that observed from January to July. Since the simulation without wet scavenging resulted in concentrations far higher than those obtained in the standard simulation and in other sensitivity experiments, the results from that simulation are plotted on a different scale (see the right y-axis of Figure 8). As discussed in the manuscript, the model simulation without convection results in larger 210Pb concentrations in the free troposphere due to the compensating effects of convective transport and scavenging. We have not reported the 7Be/210Pb ratios from sensitivity experiments since the ratio is not affected by scavenging. We have chosen to show the sensitivity tests for grid "ij-1" rather than "ij", since at the former a better comparison between the observed and simulated 210Pb and especially 7Be activities was found. Also see Figures 5-6 and their discussions.

---

## Author Comment (AC3) · 28 Nov 2016

We thank the editor for his comments. Below please find our itemized responses.

1). The authors may consider giving a brief description of what the physical and chemical processes are included in their model for the simulation of radionuclides Pb-210 and Be-7 in both the stratosphere and the troposphere, instead of just referring to the literatures (Page 10, Line 9-14).

Reply - Thanks for the suggestion. A brief description of the physical and chemical processes included in the GMI model used for the simulation of 210Pb and 7Be radionuclides in the stratosphere and the troposphere is given as follows. "In this work,

we simulate 222Rn, 210Pb, 7Be, and 10Be using a version of the GMI model with the same basic structure as described by Considine et al. (2005) and Liu et al. (2016), including parameterizations of the important tropospheric physical processes such as convection, wet scavenging, dry deposition and planetary boundary layer mixing. Meteorological data used to drive the CTM at 2° latitude by 2.5° longitude resolution, e.g., horizontal winds, convective mass fluxes and precipitation fields, are the Modern-Era Retrospective analysis for Research and Applications (MERRA) assimilated data set from the NASA Global Modeling and Assimilation Office (GMAO) (Rienecker et al., 2011)." "The flux-form semi-Lagrangian advection scheme and a convective transport algorithm from the CONVTRAN routine in NCAR CCM3 physics package are used in the model. The wet deposition scheme is that of Liu et al. (2001): it includes scavenging in wet convective updrafts, and first-order rainout and washout from both convective anvils and large-scale precipitations. The gravitational settling effect of cloud ice particles included in Liu et al. (2001) is not considered here. Dry deposition of aerosols is computed using the resistance-in-series approach."

2). Pb-210 and Be-7 were analyzed from the PM10 samples according to a description of the radionuclide measurement (Page 9, Line 5-8). I wonder if the treatment of aerosol processes in the model could affect the simulation of radionuclides. How does the model treat the uptake of radionuclides on particulate matters? Does the uncertainties in the simulation of PM10 affect the simulation results of Pb-210 and, in particular, Be-7?

Reply – The model does not specifically simulate aerosols particles to which the radionuclides attach. Instead, those aerosol particles are assumed to be ubiquitous. Now we state in Introduction: "Once produced, both radionuclides rapidly attach to ubiquitous submicron aerosol particles in the ambient air (Papastefanou and Ioannidou, 1995; Winkler et al., 1998; Gaffney et al., 2004; Ioannidou et al., 2005), and are removed from the atmosphere mainly by wet and secondarily dry deposition (Kulan et al., 2006)."

3). The treatment of Be-7 source in the model is also suggested to be described more specifically, e.g., by providing the prescribed concentrations or by giving the production rates at certain altitudes.

Reply – We have added more information on the 7Be source in the model: "Following Brost et al. (1991) and Koch et al. (1996), we used the Lal and Peters (1967) 7Be source for 1958 (solar maximum year), as it best simulated stratospheric 7Be concentrations measured from aircraft (Liu et al., 2001). The rates of 7Be production reported more recently by Usoskin and Kovaltsov (2008) broadly agree with those of Lal and Peters (1967) with slightly (about 25%) lower global production rate and will be tested in a separate model study. The Lal and Peters (1967) source is represented as a function of latitude and altitude (pressure) and does not vary with season (see Figure 1 of Koch et al., 1996)."

---

## Author Response (AR2)

**Reply to the Interactive comment of Anonymous Referee on "Processes controlling the seasonal variations of $^{210}$Pb and $^{7}$Be at the Mt. Cimone WMO-GAW global station, Italy: A model analysis" by Erika Brattich et al.**

Manuscript Ref: acp-2016-568

We thank the reviewer for the further comments. Below please find our responses.

1) Both reviewers pointed the importance of the station-based observations of precipitation at Mt Cimone for the analysis of the local Be-7 and Pb-210 measurements and the need to show these data even as supplementary material if not in Figure 4. The reviewers pointed this in order to help the reader to see how the difference between the station-based observations of precipitation at Mt Cimone and large scale grid-based precipitation data (MERRA and GPCP) could partially explain differences between the locally observed Be-7 and Pb-210 measurements and the respective modeled values. Could the authors please clarify why they prefer not showing these data?

Reply - As we have previously discussed in the manuscript (page 15, lines 18-21), it is true that the difference between the station-based observations of precipitation and large scale grid-based precipitation data (MERRA and GPCP) could contribute to the biases in our model simulated Be-7 and Pb-210 due to errors in the precipitation scavenging of radionuclides. We thus report in the text annual mean differences between the local observations and MERRA precipitation. We believe this is adequate and showing a figure does not provide additional useful information. Our corresponding text reads "Large differences between the MERRA precipitation and that locally observed at the station are instead present. While the daily mean observed 2005 precipitation is 0.81 mm, which is close to the corresponding precipitation (0.73

mm) in MERRA at the "ij" grid (i.e., a negative bias of -0.08 mm); the model bias is positive and much higher (0.31 – 1.28 mm) at adjacent grids. This bias may very well reflect again the fact that the observed surface precipitation is localized, whereas the satellite and MERRA

precipitations correspond to a much larger scale (about 200 km)."

2) The authors added in the manuscript that " However, MERRA is able to capture the summertime north-north easterly winds in the eastern Mediterranean (Aegean Sea), known as the Etesian winds, generated by thermal effects." The part of the sentence " generated by thermal effects" is a rather incomplete statement. The Etesian winds are one of the most persistent localized wind system in the world as a consequence of a sharp east–west pressure gradient manifested by large scale circulation features (low pressures over eastern

Mediterranean/Middle East and high pressure over central and southeastern Europe) (Dafka et al., Clim Dyn, 2015).

Reply - We thank the reviewer for pointing this out to us. We have revised the manuscript as follows: "
[revised manuscript text omitted]